# Crosstalk between the chloroplast protein import and SUMO systems revealed through genetic and molecular investigation in *Arabidopsis*

**Samuel James Watson[1][†], Na Li[1], Yiting Ye[2], Feijie Wu[2,3], Qihua Ling[1,2,3], R Paul Jarvis[1,3]***

[1]Department of Plant Sciences, University of Oxford, Oxford, United Kingdom; [2]National Key Laboratory of Plant Molecular Genetics, CAS Centre for Excellence in Molecular Plant Sciences, Institute of Plant Physiology and Ecology, Shanghai Institutes for Biological Sciences, Chinese Academy of Sciences, Shanghai, China; [3]Department of Biology, University of Leicester, Leicester, United Kingdom

**\*For correspondence:**
paul.jarvis@plants.ox.ac.uk

**Present address:** [†]Cambridge Institute for Therapeutic Immunology and Infectious Disease, Jeffrey Cheah Biomedical Centre, University of Cambridge, Cambridge, United Kingdom

**Abstract** The chloroplast proteome contains thousands of different proteins that are encoded by the nuclear genome. These proteins are imported into the chloroplast via the action of the TOC translocase and associated downstream systems. Our recent work has revealed that the stability of the TOC complex is dynamically regulated by the ubiquitin-dependent chloroplast-associated protein degradation pathway. Here, we demonstrate that the TOC complex is also regulated by the small ubiquitin-like modifier (SUMO) system. *Arabidopsis* mutants representing almost the entire SUMO conjugation pathway can partially suppress the phenotype of *ppi1*, a pale-yellow mutant lacking the Toc33 protein. This suppression is linked to increased abundance of TOC proteins and improvements in chloroplast development. Moreover, data from molecular and biochemical experiments support a model in which the SUMO system directly regulates TOC protein stability. Thus, we have identified a regulatory link between the SUMO system and the chloroplast protein import machinery.

## Introduction

The chloroplast is a membrane-bound organelle that houses photosynthesis in all green plants (*Jarvis and Lopez-Juez, 2013*). Chloroplasts have an unusual evolutionary history – they are the integrated descendants of a cyanobacterial ancestor that entered the eukaryotic lineage via endosymbiosis. Although chloroplasts retain small genomes, almost all of the proteins required for chloroplast development and function are now encoded by the central, nuclear genome (*Jarvis, 2008*). These proteins must be imported into the organelle after synthesis in the cytosol, and this import is mediated by the coordinate action of the TOC and TIC complexes (the translocons at the outer and inner envelope membranes of chloroplasts; *Jarvis, 2008*).

The TOC complex contains three major components: the Omp85 (outer membrane protein, 85 kDa)-related protein, Toc75, which serves as a membrane channel (*Schnell et al., 1994*; *Tranel et al., 1995*), and two GTPase-domain receptor proteins, Toc33 and Toc159 (*Hirsch et al., 1994*; *Kessler et al., 1994*; *Perry and Keegstra, 1994*; *Jarvis et al., 1998*; *Jarvis, 2008*). Toc33 and Toc159 project into the cytosol and bind incoming preproteins.

The key components of the TOC complex were identified more than two decades ago (*Hirsch et al., 1994*; *Kessler et al., 1994*; *Schnell et al., 1994*; *Tranel et al., 1995*; *Jarvis, 2008*). However,

**eLife digest** All green plants grow by converting light energy into chemical energy. They do this using a process called photosynthesis, which happens inside compartments in plant cells called chloroplasts. Chloroplasts use thousands of different proteins to make chemical energy. Some of these proteins allow the chloroplasts to absorb light energy using chlorophyll, the pigment that makes leaves green. The vast majority of these proteins are transported into the chloroplasts through a protein machine called the TOC complex. When plants lack parts of the TOC complex, their chloroplasts develop abnormally, and their leaves turn yellow.

Photosynthesis can make toxic by-products, so cells need a way to turn it off when they are under stress; for example, by lowering the number of TOC complexes on the chloroplasts. This is achieved by tagging TOC complexes with a molecule called ubiquitin, which will lead to their removal from chloroplasts, slowing photosynthesis down. It is unknown whether another, similar, molecular tag called SUMO aids in this destruction process.

To find out, Watson et al. examined a mutant of the plant *Arabidopsis thaliana*. This mutant had low levels of the TOC complex, turning its leaves pale yellow. A combination of genetic, molecular, and biochemical experiments showed that SUMO molecular tags control the levels of TOC complex on chloroplasts. Increasing the amount of SUMO in the mutant plants made their leaves turn yellower, while interfering with the genes responsible for depositing SUMO tags turned the leaves green. This implies that in plants with less SUMO tags, cells stopped destroying their TOC complexes, allowing the chloroplasts to develop better, and changing the colour of the leaves. The SUMO tagging of TOC complexes shares a lot of genetic similarities with the ubiquitin tag system.

It is possible that SUMO tags may help to control the CHLORAD pathway, which destroys TOC complexes marked with ubiquitin. Understanding this relationship, and how to influence it, could help to improve the performance of crops. The next step is to understand exactly how SUMO tags promote the destruction of the TOC complex.

the regulation of the activity and stability of the complex was, until recently, poorly understood. Major insights came from a forward genetic screen for suppressors of the pale-yellow Toc33 mutant, *ppi1* (*Ling et al., 2012*). In this screen, SP1 (SUPPRESSOR OF PPI1 LOCUS 1), a novel RING-type E3 ubiquitin ligase, was identified. A series of *sp1* mutations were shown to partially suppress the phenotypic defects of *ppi1* with respect to chlorosis, chloroplast development, and chloroplast protein import. In addition, SP1 function was shown to promote plastid interconversion events (e.g., the development of the chloroplast from its precursor organelle, the etioplast). Later work demonstrated that SP1 function is also important for abiotic stress tolerance by enabling optimization of the organellar proteome via protein import regulation (*Ling and Jarvis, 2015*). Thus, through SP1, the ubiquitin-proteasome system promotes TOC complex degradation and reconfiguration in response to developmental and/or environmental stimuli.

Ubiquitinated TOC proteins are extracted from the chloroplast outer envelope membrane and degraded in the cytosol. Recent work identified two proteins that physically associate with SP1 and promote the membrane extraction of TOC proteins (*Ling et al., 2019*). These are SP2, an Omp85-type β-barrel channel protein that was identified in the same genetic screen as SP1, and Cdc48, a well-characterized cytosolic AAA+ chaperone ATPase that provides the motive force for the extraction of proteins from the chloroplast outer envelope. The three proteins – SP1, SP2, and Cdc48 – together define a new pathway for the ubiquitination, membrane extraction, and degradation of chloroplast outer envelope proteins, which has been named chloroplast-associated protein degradation (CHLORAD). In addition to CHLORAD, there exist cytosolic ubiquitin-dependent systems that also contribute to chloroplast biogenesis by regulating the levels of unimported preproteins (*Lee et al., 2009*; *Grimmer et al., 2020*) and by controlling the stability of the Toc159 receptor prior to its integration into the outer envelope membrane (*Shanmugabalaji et al., 2018*).

The discovery of SP1 and the CHLORAD pathway demonstrated that the TOC complex is not static but, instead, can be rapidly ubiquitinated and degraded in response to developmental and environmental stimuli. To complement this work, we decided to explore whether the TOC complex is also regulated by the small ubiquitin-like modifier (SUMO) system. This work was motivated in part

by the results of a high-throughput screen for SUMO substrates in *Arabidopsis* (***Elrouby and Coupland, 2010***). This screen suggested that Toc159, a key component of the TOC complex, is a SUMO substrate. SUMOylation is intricately involved in plant development and stress adaptation, and so we were interested to determine whether the TOC complex is targeted by the SUMO system, and whether any such SUMOylation is functionally important. As crosstalk between the SUMO system and the ubiquitin-proteasome system is common, we reasoned that answering these questions might provide insights into the regulation of SP1 and the CHLORAD pathway.

To explore the relationship between the chloroplast protein import and SUMO systems, we carried out a comprehensive series of genetic, molecular, and biochemical experiments. Mutants representing most components of the *Arabidopsis* SUMO pathway were found to partially suppress the phenotype of the chlorotic Toc33 null mutant, *ppi1*, with respect to leaf chlorophyll accumulation, chloroplast development, and TOC protein abundance. Conversely, overexpression of either *SUMO1* or *SUMO3* enhanced the severity of the *ppi1* phenotype. Moreover, the E2 SUMO conjugating enzyme, SCE1, was found to physically interact with the TOC complex in bimolecular fluorescence complementation (BiFC) experiments; and TOC proteins were seen to physically associate with SUMO proteins in immunoprecipitation (IP) assays. In combination, our data conclusively demonstrate significant crosstalk between the SUMO system and the chloroplast protein import apparatus, and emphasize the complexity of the regulation of the TOC translocase.

## Results

### The E2 SUMO conjugating enzyme mutant *sce1-4* and the E3 SUMO ligase mutants *siz1-4* and *siz1-5* partially suppress the phenotype of the Toc33 mutant *ppi1*

Two key components of the CHLORAD pathway, SP1 and SP2, were identified in a forward genetic screen for suppressors of *ppi1,* an *Arabidopsis* Toc33 null mutant (***Ling et al., 2012***; ***Ling et al., 2019***). Both *sp1* and *sp2* mutants can partially suppress the *ppi1* phenotype with respect to chlorophyll accumulation, chloroplast development, and TOC protein abundance. To investigate whether the TOC complex is targeted by the SUMO system, we obtained several *Arabidopsis* SUMO system mutants, crossed them with *ppi1*, and carefully examined the phenotypes of the resulting double mutants. This reverse genetic approach was possible because the basic architecture of the *Arabidopsis* SUMO system is remarkably simple. In the SUMO pathway, the ubiquitin-like SUMO modifier protein is conjugated to substrates by the coordinated action of E1 activating enzymes, E2 conjugating enzymes, and E3 SUMO ligases. Although thousands of proteins are SUMOylated in *Arabidopsis*, there is just one known E1 SUMO activating enzyme, one known E2 SUMO conjugating enzyme, and only two known E3 SUMO ligases of canonical function (***Saracco et al., 2007***; ***Ishida et al., 2009***).

First, we analyzed *sce1-4*, a weak mutant allele of the sole E2 SUMO conjugating enzyme gene in *Arabidopsis*, which is an essential gene (***Saracco et al., 2007***). The *sce1-4* mutant shows a moderate reduction in the expression of SCE1 and in global levels of SUMOylation, but it displays no obvious visible phenotypic defects under steady-state conditions (***Saracco et al., 2007***). The *ppi1 sce1-4* double mutant was phenotypically characterized, and, intriguingly, it appeared greener than the *ppi1* single mutant (***Figure 1A***, ***Figure 1—figure supplement 1A***). This was linked to a moderate increase in leaf chlorophyll concentration (***Figure 1B***, ***Figure 1—figure supplement 1B***). Next, we asked whether the phenotypic suppression observed in *ppi1 sce1-4* was linked to changes in the development of chloroplasts. The chloroplasts of *ppi1 sce1-4* were visualized via transmission electron microscopy. Interestingly, the chloroplasts of the *ppi1 sce1-4* double mutant appeared larger and better developed than those of the *ppi1* control (***Figure 1C***). The transmission electron micrographs were quantitatively analyzed, and the *ppi1 sce1-4* chloroplasts were indeed found to be significantly larger than those of *ppi1* (***Figure 1D***), with larger, more interconnected thylakoidal granal stacks (***Figure 1E and F***).

SUMO conjugation is usually dependent on the action of E3 SUMO ligases. In *Arabidopsis*, the best characterized E3 SUMO ligase is SIZ1 (***Kurepa et al., 2003***; ***Miura et al., 2005***; ***Saracco et al., 2007***). SIZ1 is not essential, but null mutants display severely dwarfed phenotypes. In order to include SIZ1 in our genetic analysis, we obtained two new T-DNA insertion alleles and named them *siz1-4* and *siz1-5*. While both mutants were visibly similar to the published mutants (***Miura et al., 2005***;

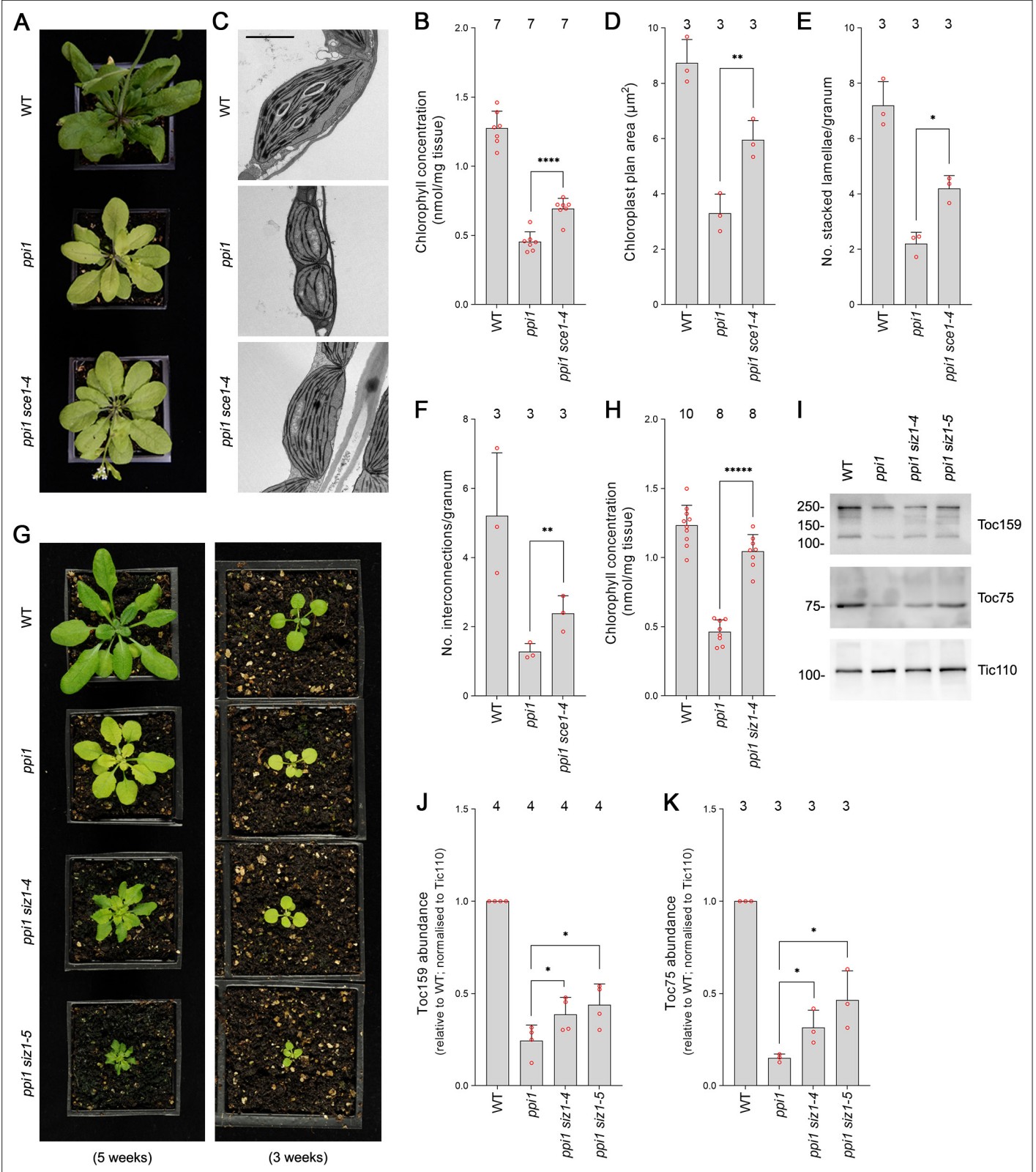

**Figure 1.** The E2 small ubiquitin-like modifier (SUMO) conjugating enzyme mutation *sce1-4* and the E3 SUMO ligase mutations *siz1-4* and *siz1-5* suppress the phenotype of the plastid protein import mutant *ppi1*. (**A**) The *ppi1 sce1-4* double mutant appeared greener than *ppi1* after approximately 5 weeks of growth on soil. (**B**) The *ppi1 sce1-4* double mutant showed enhanced accumulation of chlorophyll relative to *ppi1* after approximately 5 weeks of growth on soil. Measurements were taken from the plants shown in (**A**) on the day of photography, as well as additional similar plants. There

*Figure 1 continued on next page*

*Figure 1 continued*

were significant differences between the *ppi1* and *ppi1 sce1-4* plants (two-tailed t-test, unpaired samples, T = 6.15, p=0.000049). (**C**) Transmission electron microscopy revealed improved chloroplast development in mature rosette leaf mesophyll tissue of *ppi1 sce1-4* plants relative to *ppi1*. Plants that had been grown on soil for approximately 4 weeks were analyzed, and representative images are shown. Scale bar = 2 µm. (**D**) Chloroplast plan area was elevated in *ppi1 sce1-4* relative to *ppi1*. The transmission electron microscopy dataset was quantified. There were significant differences between the *ppi1* and *ppi1 sce1-4* plants (two-tailed t-test, unpaired samples, T = 4.65, p=0.009674). (**E, F**) Thylakoid membrane development was increased in *ppi1 sce1-4* relative to *ppi1*. The number of stacked thylakoidal lamellae per granum (**E**), and the number of stromal thylakoidal lamellae emanating from each granum (granal interconnections) (**F**), was analyzed using the transmission electron microscopy dataset. There were significant differences between the *ppi1* and *ppi1 sce1-4* plants (two-tailed t-test, unpaired samples: T = 5.53, p=0.005221 [**E**]; T = 3.38, p=0.0277 [**F**]). (**G**) The *ppi1 siz1-4* and *ppi1 siz1-5* double mutants appeared greener than *ppi1* after different periods of growth on soil. The plants were photographed after 3 weeks of growth (right panel) and then again after 5 weeks of growth (left panel). (**H**) The *ppi1 siz1-4* double mutant showed enhanced accumulation of chlorophyll relative to *ppi1* after approximately 5 weeks of growth on soil. Measurements were taken from the plants shown in (**G**) on the day of photography, as well as additional similar plants. There were significant differences between the *ppi1* and *ppi1 siz1-4* plants (two-tailed t-test, unpaired samples, T = 11.01, p<0.00001). (**I**) TOC protein accumulation was improved in *ppi1 siz1-4* and *ppi1 siz1-5* relative to *ppi1*. Analysis of the levels of Toc75 and Toc159 in *ppi1 siz1-4*, *ppi1 siz1-5*, and relevant control plants was conducted by immunoblotting. Protein samples were taken from whole seedlings that had been grown on soil for approximately 2 weeks (the plants shown in *Figure 1—figure supplement 1C* on the day of photography). Tic110, a TIC-associated protein, was included as a compartment-specific loading control (*Inaba et al., 2005*). Migration positions of standards are displayed to the left of the gel images, and sizes are indicated in kDa. Unprocessed membrane images are displayed in *Source data 1*. (**J, K**) Toc159 and Toc75 protein accumulation was improved in *ppi1 siz1-4* and *ppi1 siz1-5* relative to *ppi1*. Specific bands in (**I**) and in *Source data 1* were quantified. There were significant differences between the *ppi1* and *ppi1 siz1-4* samples (one-tailed t-test, unpaired samples: T = 2.26, p=0.032316 [**J**, Toc159]; T = 2.93, p=0.021334 [**K**, Toc75]) and between the *ppi1* and *ppi1 siz1-5* samples (one-tailed t-test, unpaired samples: T = 2.76, p=0.01639 [**J**, Toc159]; T = 3.36, p=0.014118 [**K**, Toc75]). In all bar charts, error bars indicate standard deviation from the mean, and open red circles indicate individual data points. The numbers above the graphs indicate the number of biological replicates per sample. Statistical significance is indicated as follows: *p<0.05; **p<0.01; ****p<0.0001; *****p<0.00001.

The online version of this article includes the following figure supplement(s) for figure 1:

**Figure supplement 1.** The E2 small ubiquitin-like modifier (SUMO) conjugating enzyme mutation *sce1-4* and the E3 SUMO ligase mutations *siz1-4* and *siz1-5* suppress the phenotype of the plastid protein import mutant *ppi1*.

**Figure supplement 2.** Molecular analysis of the *siz1-4* and *siz1-5* mutants.

**Figure supplement 3.** The *siz1-4* and *siz1-5* mutants display reduced global SUMOylation in response to heat shock.

---

*Liu et al., 2019*), *siz1-4* showed a milder phenotype with only moderate growth retardation when grown to maturity. We mapped the integration sites of the T-DNA insertions in these two mutants (*Figure 1—figure supplement 2A*) and showed that both display a strong reduction in *SIZ1* transcript by RT-PCR analysis (*Figure 1—figure supplement 2B*). In addition, both mutants displayed defects in global SUMOylation in response to heat shock, similar to the published alleles (*Figure 1—figure supplement 3*). The two new *siz1* mutants were crossed with *ppi1* and the resulting double mutants were phenotypically characterized. Both the *ppi1 siz1-4* and the *ppi1 siz1-5* double mutants appeared greener than the *ppi1* control (*Figure 1G*, *Figure 1—figure supplement 1C*). In addition, the double mutants showed dramatic increases in leaf chlorophyll concentration relative to *ppi1* (*Figure 1H*, *Figure 1—figure supplement 1D*). Next, we asked whether the phenotypic suppression observed in *ppi1 siz1-4* and *ppi1 siz1-5* was linked to changes in the abundance of TOC proteins. To this end, protein samples were taken from the two double mutants and relevant control plants and resolved via immunoblotting. Both double mutants displayed clear increases in the abundance of Toc159 and Toc75, two core components of the TOC complex, relative to *ppi1* (*Figure 1I, J and K*).

## The suppression effects mediated by the SUMO system mutants are specific

As discussed in the previous section, the SUMO system is encoded by a remarkably small number of genes in *Arabidopsis*. As a consequence, SUMO system mutants have highly pleiotropic molecular and physiological phenotypes. We therefore asked whether the partial suppression of *ppi1* by SUMO system mutants was specific to the *ppi1* background. We crossed *sce1-4* with *tic40-4* and *hsp93-V-1*, two TIC-complex-associated mutants. These mutants are chlorotic, due to defects in protein import across the chloroplast inner membrane, and in this respect are highly similar to *ppi1* (*Kovacheva et al., 2005*). Significantly, the resulting double mutants, *tic40-4 sce1-4* and *hsp93-V-I sce1-4*, were indistinguishable from *tic40-4* and *hsp93-V-1*, their respective single mutant controls (*Figure 2A and C*). Moreover, the double mutants did not display changes in leaf chlorophyll accumulation relative to

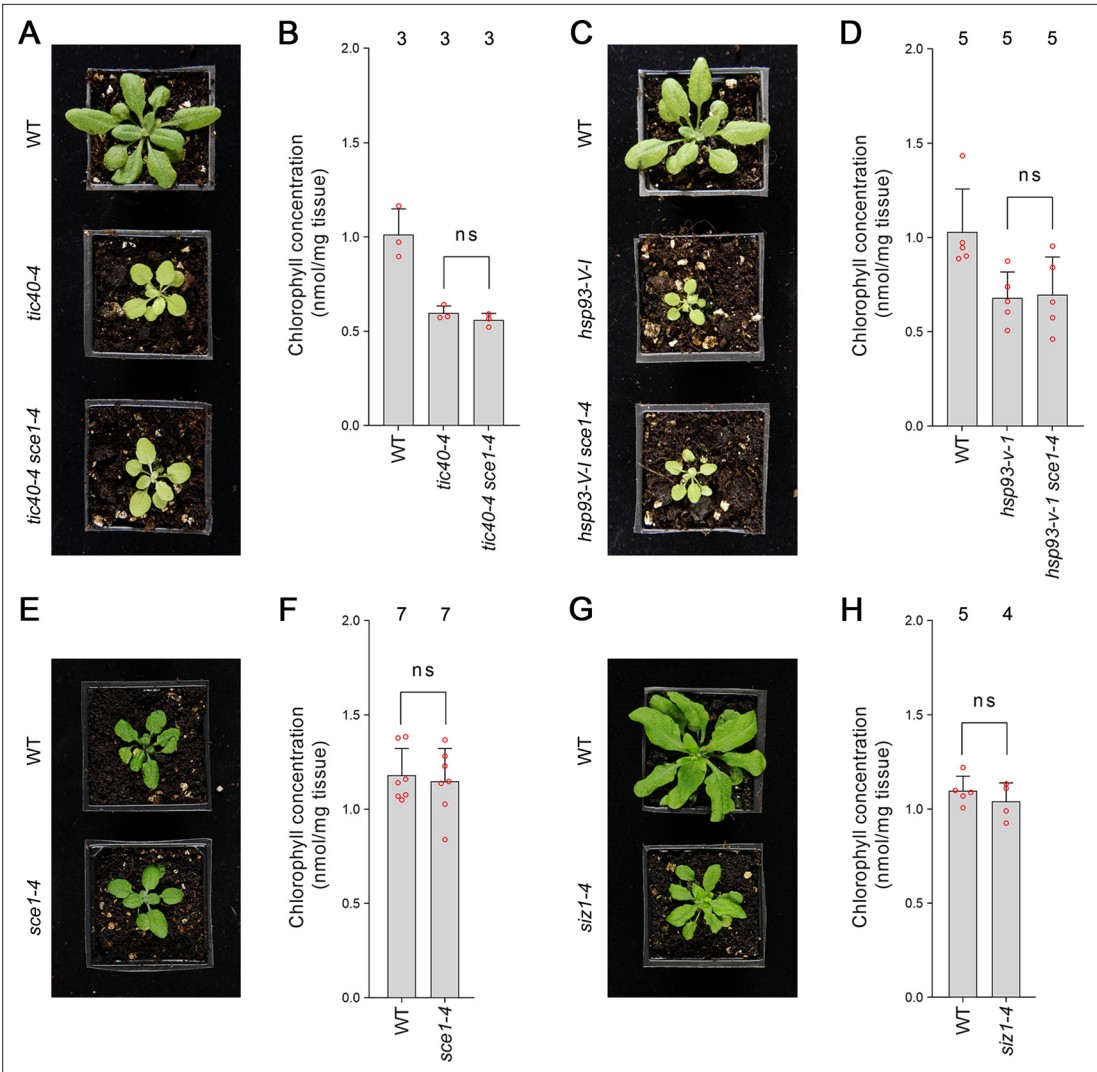

**Figure 2.** Genetic analysis reveals specificity of the suppression mediated by the *sce1-4* and *siz1-4* mutations. (**A**) The *tic40-4 sce1-4* double mutant did not appear greener than *tic40-4* after approximately 4 weeks of growth on soil. (**B**) The *tic40-4 sce1-4* double mutant did not show an enhanced accumulation of chlorophyll relative to *tic40-4* after approximately 4 weeks of growth on soil. Measurements were taken from the plants shown in (**A**) on the day of photography, as well as additional similar plants. There were no significant differences between the *tic40-4* and *tic40-4 sce1-4* plants (two-tailed t-test, unpaired samples, T = 1.25, p=0.280106). (**C**) The *hsp93-V-1 sce1-4* double mutant did not appear greener than *hsp93-V-1* after approximately 4 weeks of growth on soil. (**D**) The *hsp93-V-1 sce1-4* double mutant did not show an enhanced accumulation of chlorophyll relative to *hsp93-V-1* after approximately 4 weeks of growth on soil. Measurements were taken from the plants shown in (**C**) on the day of photography, as well as additional similar plants. There were no significant differences between the *hsp93-V-1* and *hsp93-V-1 sce1-4* plants (two-tailed t-test, unpaired samples, T = 0.18, p=0.860702). (**E**) The *sce1-4* single mutant did not appear greener than wild-type plants after approximately 4 weeks of growth on soil. (**F**) The *sce1-4* single mutant did not show an enhanced accumulation of chlorophyll relative to wild-type plants after approximately 4 weeks of growth on soil. Measurements were taken from the plants shown in (**E**) on the day of photography, as well as additional similar plants. There were no significant differences between the *sce1-4* and wild-type plants (two-tailed t-test, unpaired samples, T = 0.38, p=0.708484). (**G**) The *siz1-4* single mutant did not appear greener than wild-type plants after approximately 5 weeks of growth on soil. (**H**) The *siz1-4* single mutant did not show an enhanced accumulation of chlorophyll relative to wild-type plants after approximately 5 weeks of growth on soil. Measurements were taken from the plants shown in (**G**) on the day of photography, as well as from additional similar plants. There were no significant differences between the *siz1-4* and wild-type plants (T = 0.96, p=0.370055). In all bar charts, error bars indicate standard deviation from the mean, and open red circles indicate individual data points. The numbers above the graphs indicate the number of biological replicates per sample. Statistical significance is indicated as follows: ns: not significant.

the single mutant controls (*Figure 2B and D*). We therefore concluded that the suppression effects observed in *ppi1 sce1-4* plants were background-specific and associated with the TOC complex.

Next, we asked whether the *sce1-4* and *siz1-4* single mutants display an increase in chlorophyll concentration even in the wild-type background. However, neither mutant appeared greener than

wild-type plants (*Figure 2E and G*, *Figure 1—figure supplement 1A and C*) or displayed an increase in leaf chlorophyll concentration (*Figure 2F and H*, *Figure 1—figure supplement 1B and D*). We therefore concluded that the suppression effects mediated by the SUMO mutants were synthetic phenotypes specific to the *ppi1* background.

## BiFC analysis reveals that SCE1 physically interacts with TOC proteins

Our reverse genetic experiments revealed a genetic interaction between the E2 SUMO conjugating enzyme, SCE1, and the protein import machinery at the chloroplast outer membrane. To determine whether SCE1 directly interacts with the TOC complex, we carried out BiFC experiments in *Arabidopsis* protoplasts. The *SCE1* coding sequence was cloned into a vector that C-terminally appends the C-terminal half of YFP (cYFP) to its insert. This construct was co-expressed with various other constructs encoding TOC proteins bearing the complementary, N-terminal moiety of the YFP protein (nYFP), appended N-terminally; or with a negative control construct encoding ΔOEP7 bearing the nYFP moiety appended C-terminally. In this system, protein-protein interactions are inferred via the detection of a YFP signal, caused by the nYFP and cYFP fragments coming together to reconstitute a functional YFP protein.

Strikingly, SCE1-cYFP was found to physically associate with all tested TOC proteins – nYFP-Toc159, nYFP-Toc132, nYFP-Toc34, and nYFP-Toc33 (*Figure 3*). Moreover, these interactions were concentrated at the periphery of the chloroplasts, placing them in an appropriate subcellular context for the in situ regulation of the chloroplast protein import machinery. Conversely, SCE1-cYFP was not found to physically associate with the negative control protein ΔOEP7-nYFP. The ΔOEP7 protein comprises the transmembrane domain of plastid protein OEP7, which is sufficient to efficiently target the full-length YFP protein to the chloroplast outer membrane (*Lee et al., 2001*); thus, cYFP-ΔOEP7 serves as a location-specific negative control.

## Manipulating the expression of three SUMO isoforms alters the phenotypic severity of *ppi1*

To further explore the genetic interaction between the chloroplast protein import and SUMO systems, we crossed *ppi1* with several SUMO protein mutants. There are three major SUMO isoforms in *Arabidopsis* – SUMO1, SUMO2, and SUMO3. The *SUMO1* and *SUMO2* genes are expressed at a relatively high level throughout the plant and are largely functionally redundant (*Saracco et al., 2007*; *van den Burg et al., 2010*). In addition, they are highly similar to each other in terms of amino acid sequence (*Saracco et al., 2007*). In contrast, at steady state, *SUMO3* is expressed at a relatively low level throughout the plant, while the SUMO3 amino acid sequence is significantly divergent with respect to the other two SUMO isoforms (*van den Burg et al., 2010*).

First, we analyzed *SUMO1* and *SUMO2*. We obtained *sum1-1* and *sum2-1*, two previously characterized null mutants (*Saracco et al., 2007*), and crossed them with *ppi1*. To account for the functional redundancy between these two genes, we also sought a *ppi1 sum1-1 sum2-1* triple mutant. However, as *SUMO1* and *SUMO2* are collectively essential, *ppi1 sum1-1 sum2-1* plants that were homozygous with respect to *ppi1* and *sum2-1*, but heterozygous with respect to the *sum1-1* mutation, were selected from a segregating population. The double and triple mutants were phenotypically characterized, and all three appeared larger and greener than the *ppi1* control plants (*Figure 4A*). Moreover, the double and triple mutants showed corresponding increases in leaf chlorophyll concentration, with the triple mutant showing a larger increase than the double mutants (*Figure 4B*). These were synthetic effects as the *sum1-1*, *sum2-1*, and *sum1-1 sum2-1* single and double mutants did not appear greener than wild-type plants or show increases in chlorophyll accumulation (*Figure 4—figure supplement 1*). We therefore concluded that the *sum1-1* and *sum2-1* mutants can additively suppress the phenotype of *ppi1*.

To complement the preceding experiment, we generated transgenic plants overexpressing *SUMO1* in the *ppi1* background. The *SUMO1* coding sequence was cloned into a vector carrying a strong, constitutive promotor (cauliflower mosaic virus 35S ) upstream of the cloning site. The resulting construct was stably introduced into the *ppi1* background via *Agrobacterium*-mediated transformation. Two lines carrying a single, homozygous transgene insert were identified and taken forward for analysis. The overexpression of *SUMO1* was confirmed in both lines by RT-PCR (*Figure 4—figure supplement 2A*). Significantly, both lines displayed an accentuation of the *ppi1* phenotype: the plants

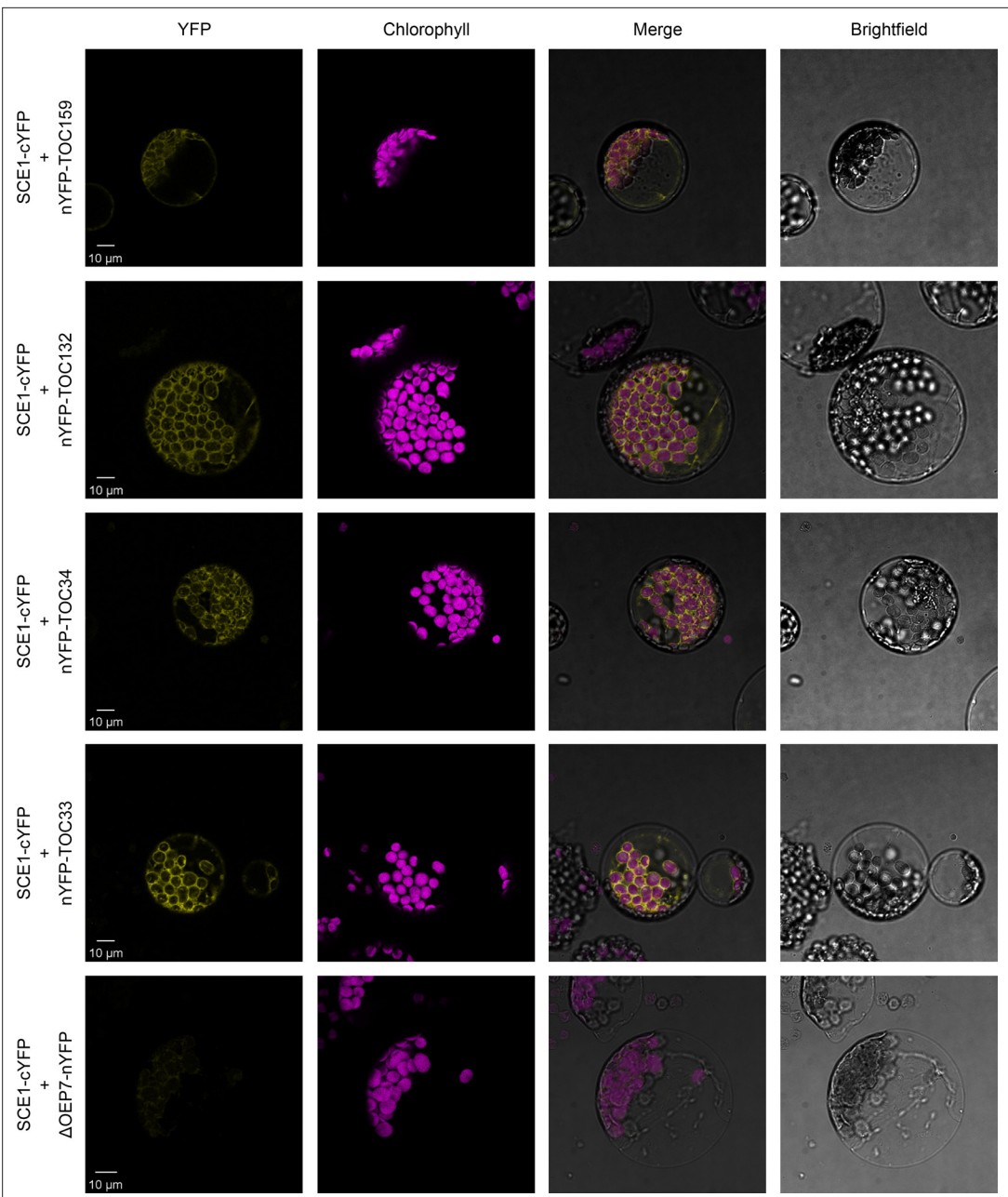

**Figure 3.** SCE1 physically interacts with TOC proteins in vivo. Bimolecular fluorescence complementation (BiFC) analysis of SCE1 protein-protein interactions was performed by imaging *Arabidopsis* protoplasts co-expressing proteins fused to complementary N-terminal (nYFP) and C-terminal (cYFP) fragments of the YFP protein, as indicated. Chlorophyll autofluorescence images were used to orientate the YFP signals in relation to the chloroplasts. SCE1 physically associated with all tested TOC proteins. The images shown are representative confocal micrographs indicating associations between SCE1 and Toc159, Toc132, Toc34, and Toc33. In contrast, SCE1 did not physically associate with ΔOEP7 (comprising the transmembrane domain of OEP7, which is sufficient to direct targeting to the chloroplast outer envelope), which served as a negative control. Scale bars = 10 µm.

were significantly smaller and paler than the *ppi1* control plants (*Figure 4C*), and showed decreases in leaf chlorophyll concentration (*Figure 4D*).

Next, we turned our attention to *SUMO3*. We obtained *sum3-1*, a previously characterized null mutant (*van den Burg et al., 2010*), and crossed it with *ppi1*. The resulting double mutant was phenotypically characterized, but it did not appear obviously different from the *ppi1* control (*Figure 4E*). Correspondingly, it did not display any clear increase in leaf chlorophyll concentration relative to *ppi1*

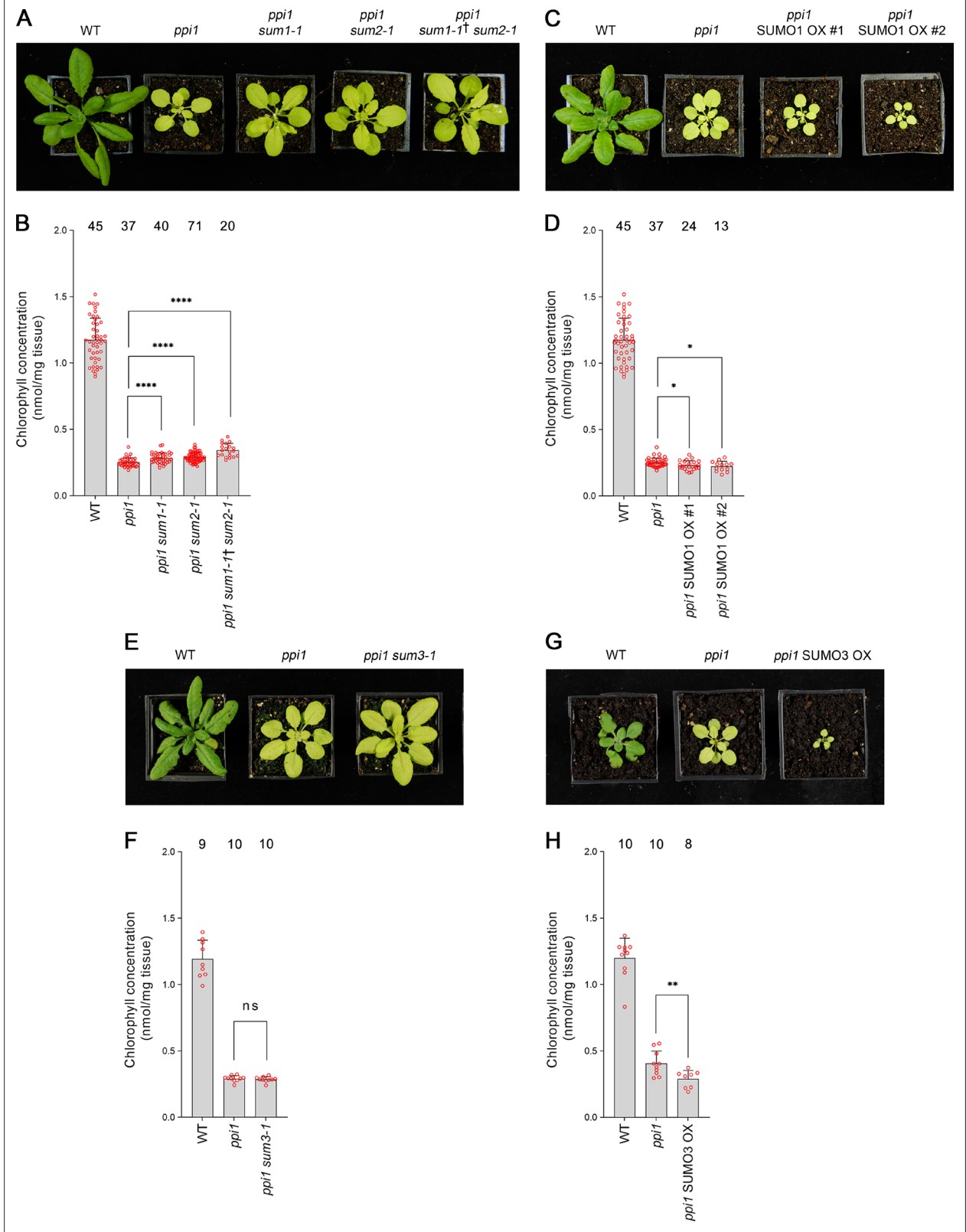

**Figure 4.** Genetic interactions between *ppi1* and the genes encoding three small ubiquitin-like modifier (SUMO) isoforms. (**A**) The *ppi1 sum1-1*, *ppi1 sum2-1*, and *ppi1 sum1-1†  sum2-1* double and triple mutants appeared greener than *ppi1* after approximately 4 weeks of growth on soil. The '†' symbol indicates that the triple mutant was heterozygous with respect to the *sum1-1* mutation. (**B**) The *ppi1 sum1-1*, *ppi1 sum2-1*, and *ppi1 sum1-1† sum2-1* double and triple mutants showed enhanced accumulation of chlorophyll relative to *ppi1* after approximately 4 weeks of growth on soil.

*Figure 4 continued on next page*

*Figure 4 continued*

Measurements were taken from the plants shown in (**A**) on the day of photography, as well as additional similar plants. There were significant differences between the samples as measured via a one-way ANOVA ($F$ = 26.21, p=6.65 × 10$^{-14}$). A post-hoc Tukey HSD test indicated that there was a significant difference between the *ppi1* and *ppi1 sum1-1* samples (p<0.00001). There were also significant differences between the *ppi1* and *ppi1 sum2-1* samples (p<0.00001), and between the *ppi1* and *ppi1 sum1-1† sum2-1* samples (p<0.00001). (**C**) The *ppi1* SUMO1 overexpression (OX) lines appeared smaller and paler than *ppi1* after approximately 4 weeks of growth on soil. (**D**) The *ppi1* SUMO1 OX lines showed reduced accumulation of chlorophyll relative to *ppi1* after approximately 4 weeks of growth on soil. Measurements were taken from the plants shown in (**C**) on the day of photography, as well as additional similar plants. There were significant differences between the *ppi1* and *ppi1* SUMO1 OX #1 plants (two-tailed t-test, unpaired samples, T = 2.27, p=0.026832), and between the *ppi1* and *ppi1* SUMO1 OX #2 plants (two-tailed t-test, unpaired samples, T = 2.49, p=0.01634). (**E**) The *ppi1 sum3-1* double mutant did not appear greener than *ppi1* after approximately 4 weeks of growth on soil. (**F**) The *ppi1 sum3-1* double mutant did not show an enhanced accumulation of chlorophyll relative to *ppi1* after approximately 4 weeks of growth on soil. Measurements were taken from the plants shown in (**E**) on the day of photography, as well as additional similar plants. There were no significant differences between the *ppi1* and *ppi1 sum3-1* plants (two-tailed t-test, unpaired samples, T = 0.54, p=0.59407). (**G**) The *ppi1* SUMO3 overexpression line appeared smaller and paler than *ppi1* after approximately 3 weeks of growth on soil. (**H**) The *ppi1* SUMO3 overexpression line showed reduced accumulation of chlorophyll relative to *ppi1* after approximately 3 weeks of growth on soil. Measurements were taken from the plants shown in (**G**) on the day of photography, as well as additional similar plants. There were significant differences between the *ppi1* SUMO3 OX plants and *ppi1* (two-tailed t-test, unpaired samples, T = 2.99, p=0.008688). In all bar charts, error bars indicate standard deviation from the mean, and open red circles indicate individual data points. The numbers above the graphs indicate the number of biological replicates per sample. Statistical significance is indicated as follows: ns: not significant; *p<0.05; **p<0.01; ****p<0.0001.

The online version of this article includes the following figure supplement(s) for figure 4:

**Figure supplement 1.** The *sum1-1*, *sum2-1*, and *sum1-1† sum2-1* single and double mutants do not show enhanced chlorophyll accumulation relative to wild-type plants.

**Figure supplement 2.** Analysis of the expression of the 35S :*SUMO1* and 35S :*SUMO3* transgenes in the selected transformants by RT-PCR.

(***Figure 4F***). To complement this experiment, we generated transgenic plants overexpressing *SUMO3* in the *ppi1* background using the approach described above, and a line carrying a single, homozygous insert was identified and taken forward for analysis. The overexpression of *SUMO3* was confirmed by RT-PCR (***Figure 4—figure supplement 2B***). Interestingly, the transgenic plants showed a striking increase in the severity of the *ppi1* phenotype: the plants were severely dwarfed and paler than the *ppi1* control (***Figure 4G***), and displayed a significant decrease in leaf chlorophyll accumulation (***Figure 4H***). These findings are particularly noteworthy when considered alongside a previous report, which explored the consequences of overexpressing *SUMO3* in wild-type plants (***van den Burg et al., 2010***). In that study, *SUMO3* overexpression was not found to alter the appearance of the transgenic plants, which implies a degree of specificity in the phenotypic accentuation observed here.

## Biochemical analysis supports SUMO action on TOC proteins in vivo

The genetic and molecular experiments described thus far strongly suggested that TOC proteins are SUMOylated. However, to our knowledge, conclusive evidence that chloroplast-resident proteins are SUMOylated is currently lacking. To investigate whether chloroplast proteins may be SUMOylated, we isolated chloroplasts from seedlings by cell fractionation and analyzed them by anti-SUMO immunoblotting. For this analysis, we employed a proven commercial antibody against SUMO1, which is one of the most abundant SUMO isoforms in *Arabidopsis* making it more tractable for analysis, and which furthermore is known to accumulate in response to heat and other stresses (***Kurepa et al., 2003***; ***van den Burg et al., 2010***). To enhance detection of SUMOylated proteins in our samples, we subjected some of the seedlings to heat shock before chloroplast isolation and/or treatment with 10 mM N-ethylmaleimide (NEM) during chloroplast isolation. NEM is a potent inhibitor of SUMO-specific proteases (***Hilgarth and Sarge, 2005***). Importantly, we detected protein SUMOylation in the isolated chloroplast samples, and this SUMOylation was increased by NEM treatment (***Figure 5—figure supplement 1***).

Next, we carried out a series of biochemical experiments to investigate whether TOC proteins may be SUMOylated. In the first of these, the *SCE1* coding sequence was cloned into a vector that appends a C-terminal YFP tag (***Karimi et al., 2002***). We confirmed that the resulting SCE1-YFP construct delivers good expression and the expected nucleocytoplasmic fluorescence pattern when transiently expressed in protoplasts (***Figure 5—figure supplement 2A***). Then, we transfected a large volume of protoplasts with the SCE1-YFP construct (or with a YFP-HA negative control construct) and performed IP using YFP-Trap magnetic beads. The samples were analyzed by immunoblotting. The YFP-HA and SCE1-YFP fusion proteins both showed robust expression and strong recovery in the

IP elutions (*Figure 5A*). Remarkably, the SCE1-YFP fusion protein was found to be associated with native Toc159 and Toc132, but not with the negative control proteins Tic110 or Tic40 (*Figure 5A*; *Kovacheva et al., 2005*; *Inaba et al., 2005* ). Conversely, YFP-HA did not associate with any of the tested proteins. Given that SCE1 is a promiscuous enzyme that associates with thousands of proteins (*Elrouby and Coupland, 2010*), and that these interactions are likely to be transient, it is remarkable that TOC co-elution was detectable in this experiment.

In the second experiment, we cloned the *SUMO1*, *SUMO2,* and *SUMO3* coding sequences into a vector that appends an N-terminal YFP tag to its insert (*Karimi et al., 2002*), a modification that previous studies have shown to be tolerated (*Ayaydin and Dasso, 2004*). All three constructs expressed well and showed the expected nucleocytoplasmic fluorescence pattern when transiently expressed in protoplasts (*Figure 5—figure supplement 2B*). The three constructs were expressed in parallel in protoplasts alongside the YFP-HA negative control construct. As in the previous experiment, the protoplasts were subjected to YFP-Trap IP, and the samples were subsequently analyzed by immunoblotting. Remarkably, all three YFP-SUMO proteins were found to physically associate with Toc159, although YFP-SUMO3 clearly bound Toc159 with the greatest affinity (*Figure 5B*, *Figure 5—figure supplement 3*). Moreover, inspection of an extended exposure of the anti-YFP blot revealed a number of higher molecular weight bands that we interpret to be SUMO adducts and indicative of the functionality of the fusions (*Figure 5—figure supplement 4*). In contrast with the SUMO fusions, the YFP-HA negative control did not associate with Toc159, and none of the four YFP fusion proteins physically associated with Tic40, a negative control protein (*Figure 5B*).

The IP experiment described above identified SUMO3 as having the highest affinity for Toc159. To extend our analysis of SUMO3 to include another TOC protein, and to more rigorously investigate the possibility of TOC protein SUMOylation, the experiment was repeated with modifications, as follows. Protoplasts were co-transfected with YFP-SUMO3 and Toc33-HA, or YFP-HA and Toc33-HA; in each case, Toc33 was transiently overexpressed to aid detection of this component and its adducts. Upon co-expression of these construct pairs, the protoplast samples were subjected to YFP-Trap IP analysis, as described earlier. In accordance with the Toc159 result (*Figure 5B*), YFP-SUMO3, but not YFP-HA, was found to physically associate with Toc33-HA (*Figure 5C*). Moreover, bands of the exact expected molecular weight for Toc33-HA bearing one or two YFP-SUMO3 moieties (75 and 114 kDa) were also detected. These bands were accompanied by a high molecular weight smear at the top of the immunoblot, which is indicative of complex, multisite or chain SUMOylation.

## Discussion

This work has revealed a genetic and molecular link between the SUMO system and the chloroplast protein import apparatus. The genetic experiments demonstrated that SUMO system mutations can suppress the phenotype of the Toc33 mutant, *ppi1*, while the molecular and biochemical experiments indicated that TOC proteins associate with key SUMO system proteins and are likely SUMOylated. Visible suppression effects observed in the *ppi1* / SUMO system double mutants were linked to improvements in chloroplast development and enhanced accumulation of key TOC proteins. Thus, our results suggest that SUMOylation acts to destabilize the TOC complex, and that when such SUMOylation is perturbed the TOC proteins are stabilized. We interpret that TOC complexes containing Toc34, Toc75, and Toc159 accumulate at higher levels in *ppi1* / SUMO system double mutants, and that this synthetically improves the double mutant phenotypes relative to the *ppi1* control. Importantly, each core TOC protein, including all of those analyzed in this study, was predicted with high probability to have one or more SUMOylation sites (*Table 1*; *Zhao et al., 2014*; *Beauclair et al., 2015*).

The *ppi1* suppression effects described here are remarkably similar to those mediated by the *sp1* and *sp2* mutations (*Ling et al., 2012*; *Ling et al., 2019*). Like *sp1* and *sp2*, SUMO system mutations can partially suppress *ppi1* with respect to chlorophyll concentration, TOC protein accumulation, and chloroplast development. This similarity suggests that SUMOylation may regulate the activity of the CHLORAD pathway. This is an attractive hypothesis as both SUMOylation and the CHLORAD pathway are activated by various forms of environmental stress (*Kurepa et al., 2003*; *Ling and Jarvis, 2015*; *Ling et al., 2019*). One possibility is that the SUMOylation of TOC proteins promotes their CHLORAD-mediated degradation. Indeed, as already noted, the ability to carry out SUMOylation is negatively correlated with the stability of TOC proteins in the context of the developed plants studied here. However, it should be kept in mind that SUMOylation can both promote and antagonize the

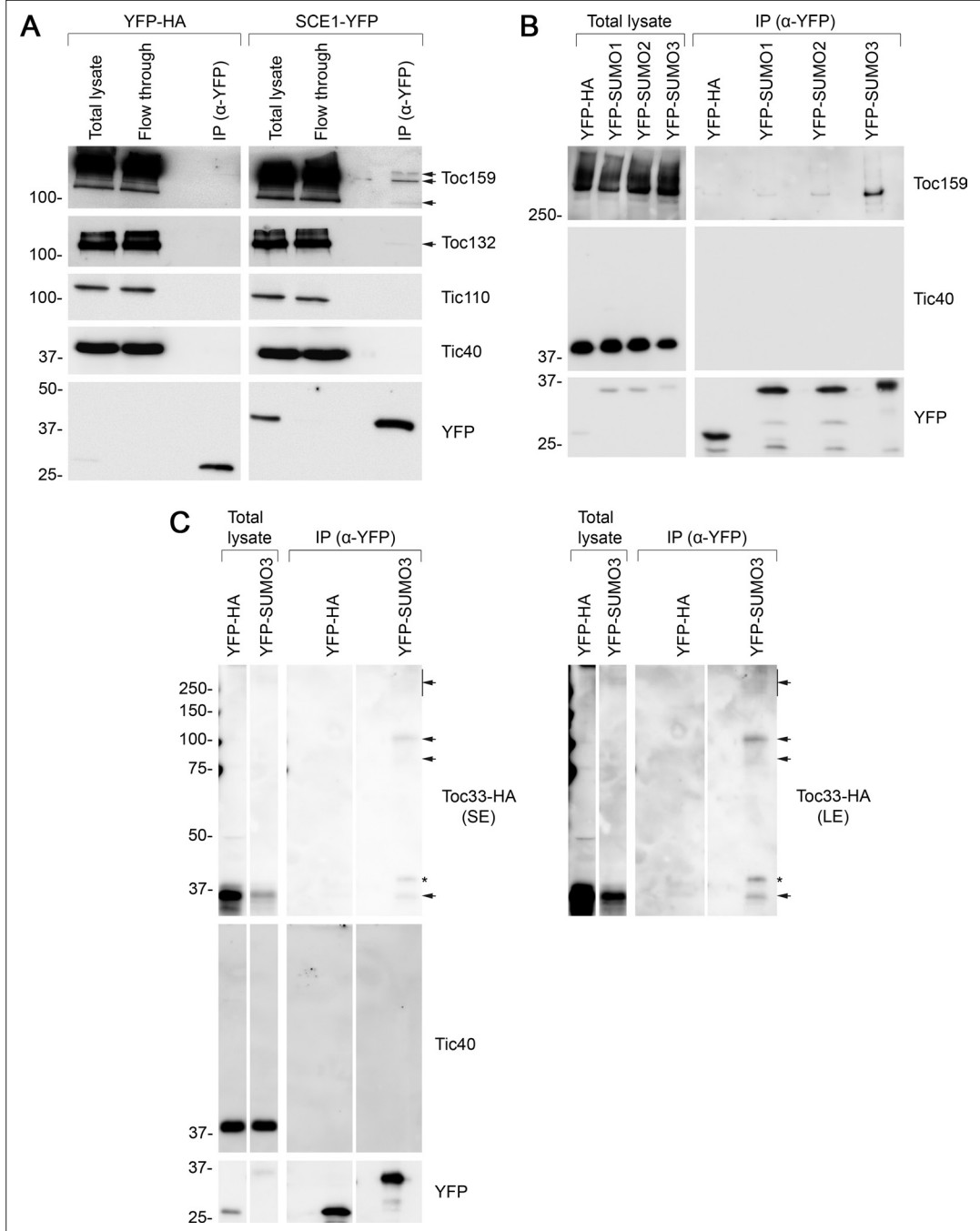

**Figure 5.** Immunoprecipitation (IP) analysis reveals that TOC proteins are small ubiquitin-like modifier (SUMO) targets. (**A**) SCE1 physically associated with native TOC proteins. *Arabidopsis* protoplasts expressing YFP-HA (left panel) or SCE1-YFP (right panel) were solubilized and subjected to IP analysis. The YFP-HA construct served as a negative control. In both cases, three samples were analyzed: the 'Total lysate' sample (total protein extract from solubilized protoplasts); the 'Flow through' sample (the total protein sample after incubation with anti-YFP beads); and the 'IP' sample (the eluted fraction of the total protein sample that bound to the anti-YFP beads). The samples were analyzed by immunoblotting, revealing that SCE1-YFP, but not the YFP-HA control, was associated with native Toc159 and Toc132 (indicated by the arrows). Neither SCE1-YFP nor YFP-HA was associated with native Tic110 or Tic40, which were included as negative control proteins. (**B**) All three SUMO isoforms physically associated with native Toc159. Protoplasts expressing YFP-HA, YFP-SUMO1, YFP-SUMO2, or YFP-SUMO3 were solubilized and subjected to IP analysis as in (**A**). In all four cases, two samples (the 'Total lysate' and the 'IP' samples) were analyzed by immunoblotting. Toc159 was resolved on an 8% acrylamide gel for 4 hr to maximize

*Figure 5 continued on next page*

*Figure 5 continued*

the resolution of high molecular weight bands. All three YFP-SUMO proteins were found to associate with native Toc159; however, YFP-SUMO3 immunoprecipitated Toc159 with the greatest efficiency. None of the four YFP fusion proteins associated with native Tic40, which served as a negative control protein. (**C**) YFP-SUMO3 physically associated with Toc33-HA and related high molecular weight species. Protoplasts co-expressing YFP-SUMO3 or YFP-HA together with Toc33-HA were solubilized and subjected to IP analysis as in (**A**). In both cases, two samples (the 'Total lysate' and 'IP' samples) were analyzed by immunoblotting. The results showed that YFP-SUMO3, but not YFP-HA, was associated with Toc33-HA (indicated by the lower arrow). Bands corresponding to the molecular weight of Toc33-HA bearing one, two, or several YFP-SUMO3 motifs were detected on the membrane (indicated by the upper arrows). The predicted molecular weight of YFP-SUMO3 is approximately 38.9 kDa. Neither YFP-SUMO3 nor YFP-HA was associated with Tic40, which served as a negative control protein. The asterisk indicates a nonspecific band. Short (SE) and long (LE) exposures are shown. Migration positions of standards are displayed to the left of the gel images, and sizes are indicated in kDa. The unprocessed membrane images are displayed in *Source data 1*.

The online version of this article includes the following figure supplement(s) for figure 5:

**Figure supplement 1.** Chloroplast resident proteins are SUMOylated.

**Figure supplement 2.** Analysis of the expression of the YFP-tagged constructs used in the immunoprecipitation experiments by confocal microscopy.

**Figure supplement 3.** All three small ubiquitin-like modifier (SUMO) isoforms physically associated with native Toc159, with SUMO3 showing the strongest association.

**Figure supplement 4.** All three YFP-SUMO probes are conjugation-competent.

effects of ubiquitination in different situations (*Desterro et al., 1998*; *Ahner et al., 2013*; *Liebelt and Vertegaal, 2016*); and so our results do not preclude the possibility that SUMOylation may have different consequences for chloroplast biogenesis in other contexts. The Toc159 receptor is regulated by SP1 when integrated into the outer envelope membrane (*Ling et al., 2012*; *Ling et al., 2019*), but by a different E3 ligase when it exists as a cytosolic precursor during the earliest stages of development before germination (*Shanmugabalaji et al., 2018*). Thus, regulation by SUMOylation might be similarly different in these two distinct developmental contexts.

The precise mechanisms underpinning the observed negative regulation of the TOC apparatus by SUMOylation are currently unknown. One possibility is that the SUMOylation of TOC proteins promotes their association with SP1. SUMOylation can modify protein-protein interactions, and some RING-type E3 ubiquitin ligases specifically recognize SUMOylated substrates (*Sriramachandran and Dohmen, 2014*). However, these SUMO-targeted ubiquitin ligases (STUbLs) typically contain SUMO-interacting motifs (SIMs), which guide the ligases to SUMO proteins conjugated to their substrates, and these are not apparent in SP1 (data not shown; *Zhao et al., 2014*). However, SP1 forms a complex with SP2 and very likely additional cofactors, and these could hypothetically provide a SUMO binding interface. Another possibility is that SUMOylation could be involved in the recruitment of Cdc48 from the cytosol. Two important Cdc48 cofactors are Ufd1 and Npl4, and the former contains a SIM, which can guide Cdc48 to SUMOylated proteins (*Nie et al., 2012*; *Baek et al., 2013*). Moreover, the SUMO-mediated recruitment of Cdc48 has important roles in the maintenance of genome stability in yeast (*Bergink et al., 2013*).

The biochemical experiments described in this article indicate that, of the three SUMO isoforms tested, SUMO3 binds TOC proteins with the highest affinity. However, there is an apparent incongruence between the results of these experiments and the results of the genetic experiments. While the *sum1-1* and *sum2-1* mutants were found to additively suppress *ppi1*, the *sum3-1* mutant did not suppress *ppi1*. At face value, this seems puzzling; however, it can be explained by the relative abundance of the three SUMO proteins in planta. SUMO1 and SUMO2 are highly abundant relative to SUMO3, which is, at steady state, very weakly abundant (*van den Burg et al., 2010*). The IP data shown in *Figure 5B* indicated that SUMO1 and SUMO2 can weakly interact with Toc159, and so it is likely that these two isoforms can compensate for the loss of SUMO3 in the *sum3-1* mutant. Although SUMO3 associates with TOC proteins with the highest affinity, the higher abundance of the other two SUMO proteins may facilitate such compensation. It is also noteworthy that, when overexpressed, SUMO3 accentuates the *ppi1* phenotype to a far greater extent than does SUMO1.

**Table 1.** Bioinformatic analysis predicts that the core TOC proteins in *Arabidopsis* contain SUMOylation sites and small ubiquitin-like modifier (SUMO) interaction motifs.

The GPS-SUMO algorithm was applied to the amino acid sequences of Toc159, Toc132, Toc120, Toc90, Toc75, Toc33, and Toc34 using the 'high stringency' setting, and the results generated are shown in columns 2–4. 'Consensus' sites fall within canonical SUMO site motifs: $\phi$-K-X-E (where $\phi$ indicates a hydrophobic amino acid, and X indicates any amino acid residue). 'Non-consensus' sites do not fall within canonical SUMO site motifs; analysis shows that ~40% of SUMOylation may occur at non-consensus sites (*Zhao et al., 2014*). 'SUMO interaction' sites are predicted to mediate the non-covalent interaction between proteins and SUMO peptides. The JASSA algorithm was also applied to the amino acid sequences using the 'high cutoff' setting (see column 5). aa denotes amino acids.

| Protein name (identifier and length) | Position (aa) | p-Value | Type | Also predicted by JASSA (high stringency)? |
|---|---|---|---|---|
| **atToc159** | 95 | 0.021 | Consensus | No |
| (At4g02510) | 106 | 0.034 | Consensus | Yes |
| 1503 aa | 126 | 0.032 | Consensus | Yes |
| | 144 | 0.02 | Consensus | Yes |
| | 151 | 0.02 | Consensus | No |
| | 246–250 | 0.005 | SUMO interaction | Yes |
| | 408–412 | 0.009 | SUMO interaction | Yes |
| | 486–490 | 0.009 | SUMO interaction | Yes |
| | 498 | 0.022 | Consensus | No |
| | 502 | 0.049 | Non-consensus | No |
| | 539 | 0.026 | Consensus | No |
| | 1,300 | 0.049 | Non-consensus | No |
| | 1,370 | 0.01 | Consensus | Yes |
| **atToc132** | 30 | 0.006 | Consensus | Yes |
| (At2g16640) | 66 | 0.036 | Consensus | Yes |
| 1206 aa | 352 | 0.002 | Consensus | No |
| | 895 | 0.005 | Consensus | No |
| | 1077 | 0.014 | Consensus | No |
| **atToc120** | 52 | 0.008 | Consensus | No |
| (At3g16620) | 57 | 0.031 | Consensus | No |
| 1084 aa | 209 | 0.05 | Non-consensus | No |
| | 777 | 0.006 | Consensus | No |
| | 959 | 0.013 | Consensus | No |
| **atToc90** | 191 | 0.027 | Consensus | Yes |
| (At5g20300) | 481 | 0.017 | Consensus | No |
| 793 aa | 711 | 0.02 | Consensus | No |
| | 786 | 0.042 | Non-consensus | Yes |

*Table 1 continued on next page*

*Table 1 continued*

| Protein name (identifier and length) | Position (aa) | p-Value | Type | Also predicted by JASSA (high stringency)? |
|---|---|---|---|---|
| **atToc75** | 434 | 0.049 | Non-consensus | No |
| (At3g46740) | 513 | 0.029 | Consensus | No |
| 818 aa | | | | |
| **atToc33** | 291 | 0.044 | Non-consensus | No |
| (At1g02280) | | | | |
| 297 aa | | | | |
| **atToc34** | 290 | 0.026 | Consensus | No |
| (At5g05000) 313 aa | 298 | 0.043 | Non-consensus | Yes |

It is now well established that the regulation of chloroplast protein import has critical roles in plant development and stress acclimation (*Sowden et al., 2018*; *Watson et al., 2018*). Here, we demonstrate regulatory crosstalk between the SUMO system and the chloroplast protein import machinery, and present results that are consistent with a model in which SUMOylation modulates the activity or effects of the CHLORAD pathway. The precise nature of the links between these two critically important control systems will be the subject of future investigation.

# Materials and methods

## Key resources table

| Reagent type (species) or resource | Designation | Source or reference | Identifiers | Additional information |
|---|---|---|---|---|
| Genetic reagent (*Arabidopsis thaliana*) | Col-0 | NASC | NASC (RRID:SCR_004576) | |
| Genetic reagent (*Arabidopsis thaliana*) | *ppi1-1* (Col-0) | NASC | NASC_ID:N2107726 | |
| Genetic reagent (*Arabidopsis thaliana*) | *sce1-4* (Col-0) | NASC | NASC_ID:N506164 | |
| Genetic reagent (*Arabidopsis thaliana*) | *siz1-2* (Col-0) | NASC | NASC_ID:N6559 | |
| Genetic reagent (*Arabidopsis thaliana*) | *siz1-3* (Col-0) | NASC | NASC_ID:N6560 | |
| Genetic reagent (*Arabidopsis thaliana*) | *siz1-4* (Col-0) | NASC | NASC_ID:N862769 | |
| Genetic reagent (*Arabidopsis thaliana*) | *siz1-5* (Col-0) | NASC | NASC_ID:N611280 | |
| Genetic reagent (*Arabidopsis thaliana*) | *sum1-1* (Col-0) | NASC | NASC_ID:N872916 | |
| Genetic reagent (*Arabidopsis thaliana*) | *sum2-1* (Col-0) | NASC | NASC_ID:N629775 | |
| Genetic reagent (*Arabidopsis thaliana*) | *sum3-1* (Col-0) | NASC | NASC_ID:N623673 | |

*Continued on next page*

*Continued*

| Reagent type (species) or resource | Designation | Source or reference | Identifiers | Additional information |
|---|---|---|---|---|
| Genetic reagent (*Arabidopsis thaliana*) | *ppi1* SUMO1 OX #1 Col-0 | This study | | |
| Genetic reagent (*Arabidopsis thaliana*) | *ppi1* SUMO1 OX #2 Col-0 | This study | | |
| Genetic reagent (*Arabidopsis thaliana*) | *ppi1* SUMO3 OX #1 Col-0 | This study | | |
| Antibody | Anti-SUMO1 (rabbit polyclonal) | Abcam | Ab5316 | WB (1:1000) |
| Antibody | Anti-Toc75-III (rabbit polyclonal) | doi:10.1111/j.1365–313 X.2011.04551.x | Custom-made | WB (1:500) |
| Antibody | Anti-Toc159 (rabbit polyclonal) | doi:10.1038/35003214 | Custom-made | WB (1:2500) |
| Antibody | Anti-Toc132 (rabbit polyclonal) | doi:10.1126/science.1225053 | Custom-made | WB (1:1000) |
| Antibody | Anti-Toc33 (rabbit polyclonal) | doi:10.1111/j.1365–313 X.2011.04551.x | Custom-made | WB (1:1000) |
| Antibody | Anti-Tic110 (rabbit polyclonal) | doi:10.1111/j.1365–313 X.2010.04242.x | Custom-made | WB (1:1000) |
| Antibody | Anti-Tic40 (rabbit polyclonal) | doi:10.1111/j.1365–313 X.2011.04551.x | Custom-made | WB (1:10,000) |
| Antibody | Anti-GFP (rabbit polyclonal) | Sigma | SAB4301138 | WB (1:1000) |
| Antibody | Goat anti-rabbit IgG HRP (goat polyclonal) | Sigma | 12-348 | WB (1:10,000) |
| Antibody | Goat anti-rabbit IgG alkaline phosphatase (goat polyclonal) | Sigma | A3687 | WB (1:10,000) |
| Recombinant DNA reagent | pSAT4(A)-nEYFP-N1 | doi:10.1007/s11103-005-0340-5 doi:10.1016/j.jmb.2006.08.017 | | Kind gift from the Gelvin lab (Purdue University) |
| Recombinant DNA reagent | pSAT4-cEYFP-C1-B | doi:10.1007/s11103-005-0340-5 doi:10.1016/j.jmb.2006.08.017 | | Kind gift from the Gelvin lab (Purdue University) |
| Recombinant DNA reagent | pSAT4(A)-cEYFP-N1 | doi:10.1007/s11103-005-0340-5 doi:10.1016/j.jmb.2006.08.017 | | Kind gift from the Gelvin lab (Purdue University) |
| Recombinant DNA reagent | pDONR201 | Invitrogen | RRID:Addgene#2392 | |
| Recombinant DNA reagent | pDONR207 | Invitrogen | RRID:Addgene#2393 | |
| Recombinant DNA reagent | pH2GW7 | doi:10.1016/s1360-1385 (02)02251–3 doi:10.1016/j.tplants.2005.01.008 | | VIB-UGent Center for Plant Systems Biology |
| Recombinant DNA reagent | p2GWY7 | doi:10.1016/s1360-1385 (02)02251–3 doi: 10.1016/j.tplants.2005.01.008 | | VIB-UGent Center for Plant Systems Biology |
| Recombinant DNA reagent | p2YGW7 | doi:10.1016/s1360-1385 (02)02251–3 doi:10.1016/j.tplants.2005.01.008 | | VIB-UGent Center for Plant Systems Biology |
| Chemical compound, drug | N-ethylmaleimide | Merck | E3876 | |
| Software, algorithm | GPS-SUMO | http://sumosp.biocuckoo.org/online.php | | |
| Software, algorithm | JASSA | http://www.jassa.fr/index.php | | |

## Plant material and growth conditions

All *Arabidopsis thaliana* plants used in this work were of the Columbia-0 (Col-0) ecotype. The mutants used in most of the analyses (*ppi1*, *sce1-4*, *sum1-1*, *sum2-1*, *sum3-1*, *hsp93-V-I*, *tic40-4*) have been described previously (*Jarvis et al., 1998*; *Kovacheva et al., 2005*; *Saracco et al., 2007*; *van den Burg et al., 2010*). The *siz1-4* (SAIL_805_A10) and *siz1-5* (SALK_111280) mutants were obtained from the Salk Institute Genomic Analysis Laboratory (SIGnAL) (*Alonso et al., 2003*) via the Nottingham *Arabidopsis* Stock Centre (NASC). Each line was verified via PCR genotyping (see *Table 2* for primer sequences) and phenotypic analysis (including the double and triple mutants). The positions of the T-DNA insertions were mapped via PCR. The following primer pairs were used to generate diagnostic amplicons from genomic DNA: LB1 and Siz1-Seq-1R (for mapping *siz1-4*), and LBb1 and Siz1-Seq-3R (for mapping *siz1-5*) (see *Table 2D and E*). The amplicons were sequenced and the positions of the T-DNA insertions inferred from the sequence data.

In most experiments, plants were grown on soil (80% [v/v] compost [Levington M2], 20% [v/v] vermiculite [Sinclair Pro, medium particle size]). However, where plants were grown for selection of transformants or for chloroplast isolation, seeds were surface sterilized and sown on petri plates containing Murashige–Skoog (MS) agar medium. The plates were stored at 4 °C for 48 hr before being transferred to a growth chamber. Both soil-grown and plate-grown plants were kept in a growth chamber (Percival Scientific) under long-day conditions (16 hr light, 8 hr dark). The light intensity was approximately 120 µE m$^{-2}$ s$^{-1}$, the temperature was held constant at 20 °C, and the humidity was held constant at approximately 60% (relative humidity).

## Chlorophyll measurements

Chlorophyll measurements were taken from mature rosette leaves in each instance. A handheld Konica-Minolta SPAD-502 meter was used to take each measurement, and the raw values were converted into chlorophyll concentration values (nmol/mg tissue) via published calibration equations (*Ling et al., 2011*).

## Chloroplast isolation and protein extraction

Chloroplasts were isolated from 14-day-old, plate-grown seedlings as described previously (*Flores-Pérez and Jarvis, 2017*). Some of the seedlings were heat-shocked immediately prior to chloroplast isolation. To do this, the plates containing the seedlings were wrapped in clingfilm and placed into a water bath (42 °C for 1 hr, followed by a 1 hr recovery period at 22 °C). Protein samples were prepared from the isolated chloroplasts by extraction using SDS-PAGE sample buffer, as well as from whole 14-day-old seedlings as previously described (*Kovacheva et al., 2005*). In some cases, the samples were treated with 10 mM NEM (*Hilgarth and Sarge, 2005*); this was added directly to the protein extraction buffer (whole seedling samples) or to the chloroplast isolation buffer following polytron homogenization and all subsequent buffers (chloroplast samples).

## Plasmid constructs

The constructs used in the BiFC experiments were generated as follows. The coding sequences of *SCE1*, *SIZ1*, *TOC159*, *TOC132*, *TOC34,* and *TOC33* were PCR amplified from wild-type cDNA (see *Table 2* for primer sequences). In the case of *ΔOEP7*, the first 105 base pairs of the *OEP7* coding sequence were amplified; this encodes a truncated sequence, which is sufficient to efficiently target the full-length YFP protein to the chloroplast outer envelope membrane (*Lee et al., 2001*). The inserts were cloned into one of the following complementary vectors: pSAT4(A)-cEYFP-N1 (*SCE1*), pSAT4-nEYFP-C1 (*TOC159*, *TOC132*, *TOC34*, *TOC33*), or pSAT4(A)-nEYFP-N1 (*ΔOEP7*), which were described previously (*Tzfira et al., 2005*; *Citovsky et al., 2006*).

The constructs used in the IP experiments were generated as follows. The coding sequences of *SCE1*, *SUMO1*, *SUMO2,* and *SUMO3* were PCR amplified from wild-type cDNA using primers bearing 5' attB1 and attB2 adaptor sequences (see *Table 2* for primer sequences). The amplicons were then cloned into pDONR207 (Invitrogen), a Gateway entry vector. The inserts from the resulting entry clones were then transferred to one of two destination vectors: p2GWY7 (*SCE1*) or p2YGW7 (*SUMO1*, *SUMO2*, *SUMO3*); the former appends a C-terminal YFP tag to its insert, and the latter appends an N-terminal YFP tag to its insert (*Karimi et al., 2002*; *Karimi et al., 2005*). The Toc33-HA and YFP-HA constructs have been described previously (*Ling et al., 2019*).

**Table 2.** Primers used during the course of this study.

(A)Primers used in restriction cloning procedures

| Primer name | Sequence | Used to generate … |
|---|---|---|
| SCE1 F (HindIII) | AA<u>AAGCTT</u>ATGGCTAGTGGAATCGCTC | pSAT4(A)-nEYFP-N1 SCE1 |
| SCE1 R (EcoRI) | AA<u>GAATTC</u>GACAAGAGCAGGATACTGCTTG | pSAT4(A)-nEYFP-N1 SCE1 |
| Toc159-5 F (EcoRI) | AA<u>GAATTC</u>AATGGACTCAAAGTCGGTT | pSAT4-cEYFP-C1-B Toc159 |
| Toc132-5 F (XhoI) | AA<u>CTCGAGC</u>TATG GGAGATGGGACTGAG | pSAT4-cEYFP-C1-B Toc159 |
| Toc132-3 R (SmaI) | AA<u>CCCGGG</u>TCATTGTCCATATTGCGT | pSAT4-cEYFP-C1-B Toc132 |
| Toc33-5 F (HindIII) | AG<u>AAGCTT</u>CGATGGGGTCTCTCGTTCGT | pSAT4-cEYFP-C1-B Toc132 |
| Toc33-3 R (XbaI) | AA<u>TCTAGA</u>TTAAAGTGGCTTTCCACT | pSAT4-cEYFP-C1-B Toc33 |
| Toc34-5 F (HindIII) | AG<u>AAGCTT</u>CGATGGCAGCTTTGCAAACG | pSAT4-cEYFP-C1-B Toc34 |
| Toc34-3 R (XbaI) | AA<u>TCTAGA</u>TCAAGACCTTCGACTTGC | pSAT4-cEYFP-C1-B Toc34 |
| OEP7 F (XhoI) | <u>CTCGAG</u>ATGGGAAAAACTTCGGGA | pSAT4(A)-cEYFP-N1 ΔOEP7 |
| OEP7-35 R (KpnI) | <u>GGTACCG</u>GAATTTATCGAGGAAAGG | pSAT4(A)-cEYFP-N1 ΔOEP7 |

(B)Primers used in Gateway cloning procedures

| | | |
|---|---|---|
| SCE1 Gateway F | <u>GGGGACAAGTTTGTACAAAAAAGCAGGCTCC</u>ATGGCTAGTGGAATCGCTC | p2GWY7 SCE1 |
| SCE1 Gateway R | <u>GGGGACCACTTTGTACAAGAAAGCTGGGTT</u>GACAAGAGCAGGATACTGC | p2GWY7 SCE1 |
| SUMO1 Gateway F | <u>GGGGACAAGTTTGTACAAAAAAGCAGGCTCC</u>TCTGCAAACCAGGAGGAAGACAAG | p2YGW7 SUMO1 |
| SUMO1 Gateway R | <u>GGGGACCACTTTGTACAAGAAAGCTGGGTTT</u>CAGGCCGTAGCACCACC | p2YGWY SUMO1 |
| SUMO2 Gateway F | <u>GGGGACAAGTTTGTACAAAAAAGCAGGCTCC</u>TCTGCTACTCCGGAAGAAGAC | p2YGW7 SUMO2 |
| SUMO2 Gateway R | <u>GGGGACCACTTTGTACAAGAAAGCTGGGTT</u>CTAAAAGCAGAAGAGCTTCAGGCC | p2YGW7 SUMO2 |

*Table 2 continued on next page*

*Table 2 continued*

(A)Primers used in restriction cloning procedures

| SUMO3 Gateway F | GSGGGACAAGTTTGTACAAAAAAGCAGGCTCCTCTAACCCTCAAGATGACAAGCCC | p2YGW7 SUMO3 |
|---|---|---|
| SUMO3 Gateway R | GGGGACCACTTTGTACAAGAAAGCTGGGTTTTAAAGCCCATTATGATCGAAAAGC | p2YGW7 SUMO3 |

(C)Primers used in RT-PCR experiments

| SUMO1 F(2) | AAAAAGCAGGCTCCACAAAAGCCACGGCCAATTAG | *SUMO1* |
|---|---|---|
| SUMO1 R(2) | AGAAAGCTGGGTTCCATTCATATCACACACAAGCCC | *SUMO1* |
| SUMO3 F | ACAGACTGGAGTTTTTGTTTC | *SUMO3* |
| SUMO3 R | CTCATGAGTCATTTACACACACG | *SUMO3* |
| eIF4E1 F | AAGATTTGAGAGGTTTCAAGCGGTGTAAG | *eIF4E1* |
| eIF4E1 R | AAACAATGGCGGTAGAAGACACTC | *eIF4E1* |
| Siz1-HindIII-F | AAAAGCTTATGGATTTGGAAGCTAATTGTAAG | *SIZ1 (for siz1-4)* |
| Siz1-seq-1R | TCTGCATTGTGCTTGCAC | *SIZ1 (for siz1-4)* |
| Siz1-F | GGATTATCTTCCAGTAATAGGCAAG | *SIZ1 (for siz1-5)* |
| Siz1-R | CCCGACTGAGCTGAAGCATC | *SIZ1 (for siz1-5)* |

(D)Primers used to genotype mutants

| Primer name | Sequence | Used to genotype… |
|---|---|---|
| **SUMO1 F(2)** | AAAAAGCAGGCTCCACAAAAGCCACGGCCAATTAG | *sum1-1* |
| SUMO1 R(2) | AGAAAGCTGGGTTCCATTCATATCACACACAAGCCC | *sum1-1* |
| SUMO2 F | CGTTGTTGGTACTTGGTTGG | *sum2-1* |
| SUMO2 R | CAAAACTCTAAACTGGTCGG | *sum2-1* |
| SUMO3 F | ACAGACTGGAGTTTTTGTTTC | *sum3-1* |
| SUMO3 R | CTCATGAGTCATTTACACACACG | *sum3-1* |
| SCE1 F | CGCCGCGAAATCTGGACC | *sce1-4* |
| SCE1 R | TTCCTCTCTTCAGCTAAACG | *sce1-4* |
| SIZ1 F(2) | GCAAACAGGGAAAGAAGCAGG | *siz1-4* |
| SIZ1 R(2) | CATTGAGTCTGTTTCTAGCG | *siz1-4* |
| LBb1 | GCGTGGACCGCTTGCTGCAACT | SALK lines (left border) |
| LB1 | GCCTTTTCAGAAATGGATAAATAGCCTTGCTTCC | SAIL lines (left border) |

(E)Primers used to map the*siz1-4*and*siz1-5*T-DNA insertions

| Primer name | Sequence | Used to map … |
|---|---|---|

*Table 2 continued on next page*

*Table 2 continued*

(A)Primers used in restriction cloning procedures

| Siz1-Seq-1R | TCTGCATTGTGCTTGCAC | | *siz1-4* T-DNA insertion |
|---|---|---|---|
| Siz1-Seq-3R | | TGACAACCACTGTATGCAGG | *siz1-5* T-DNA insertion |

Underlining indicates restriction sites (A and C), or attB1 and attB2 recombination sequences (B).

The constructs used to generate transgenic plants were generated as follows. The coding sequences of *SUMO1* and *SUMO3* were PCR amplified from wild-type cDNA using primers bearing 5′ attB1 and attB2 adaptor sequences (see *Table 2* for primer sequences). The inserts were then cloned into pDONR201 (Invitrogen), a Gateway entry vector. The inserts from the resulting entry clones were then transferred to the pH2GW7 binary destination vector (*Karimi et al., 2002*; *Karimi et al., 2005*).

## Transient expression assays

Protoplasts were isolated from mature rosette leaves of wild-type *Arabidopsis* plants and transfected in accordance with an established method (*Wu et al., 2009*; *Ling et al., 2012*). In the BiFC experiments, 100 µL protoplast suspension (containing approximately $10^5$ protoplasts) was transfected with 5 µg plasmid DNA; in the IP experiments, 600 µL protoplast suspension (containing approximately 6 × $10^5$ protoplasts) was transfected with 30 µg plasmid DNA. In both cases, the samples were analyzed after 15–18 hr.

## Stable plant transformation

Transgenic lines carrying the SUMO1 OX or SUMO3 OX constructs were generated via *Agrobacterium*-mediated floral dip transformation (*Clough and Bent, 1998*). Transformed plants ($T_1$ generation) were selected on MS medium containing phosphinothricin. Multiple $T_2$ families were analyzed in each case, and lines bearing a single T-DNA insertion were taken forward for further analysis. Transgene expression was analyzed by semi-quantitative RT-PCR as described previously (*Kasmati et al., 2011*; see *Table 2* for primer sequences).

## Transmission electron microscopy

Transmission electron micrographs were recorded using mature rosette leaves as previously described (*Huang et al., 2011*). Images were taken from three biological replicates (different leaves from different individual plants), and at least 10 images were taken per replicate. The images were analyzed using ImageJ (*Schneider et al., 2012*). The freehand tool was used to measure the plan area of the chloroplasts. For this, between 9 and 28 chloroplasts were analyzed for each biological replicate (i.e., for each plant), and then an average value for each replicate was calculated and used for statistical comparisons. The analysis of chloroplast ultrastructure was performed as in previous work (*Huang et al., 2011*). For this, between 3 and 8 chloroplasts were analyzed per biological replicate, and the data were processed as above.

## BiFC experiments

The BiFC experiments were carried out as described previously (*Ling et al., 2019*). Protoplasts were co-transfected with two constructs encoding fusion proteins bearing complementary fragments of the YFP protein (nYFP and cYFP; *Citovsky et al., 2006*). After transfection and overnight incubation, the protoplasts were imaged using a Leica TCS SP5 laser scanning confocal microscope equipped with a Leica HC Plan Apochromat CS2 63.0× UV water immersion lens with a numerical aperture (N.A.) of 1.2. YFP was excited with an argon-ion laser at 514 nm, selected using an acousto-optic tuneable filter (AOTF), and was detected using a 525–600 nm bandpass filter and a photomultiplier. Chlorophyll fluorescence was simultaneously excited with 514 nm excitation and detected with a 680–700 nm bandpass filter using a photomultiplier. Images were collected in 8-bit resolution with the pinhole set at 111.5 µm (1 Airy Unit), using 16-line averaging and a scan speed of 400 Hz. The image size was

512 × 512 pixels, with an (x,y) pixel size of 0.239 μm. Images were processed in the Leica Application Suite (LAS) software.

### Immunoblotting and immunoprecipitation

Protein extraction and immunoblotting were performed as described previously (*Kovacheva et al., 2005*). Total protein samples were extracted from 50 mg of intact, pooled seedlings after 2 weeks of growth. To detect proteins, we used an anti-SUMO1 antibody (Ab5316, Abcam), an anti-Toc75-III antibody (*Kasmati et al., 2011*), an anti-Toc159 antibody (*Bauer et al., 2000*), an anti-Toc132 antibody (*Ling et al., 2012*), an anti-Toc33 antibody (*Kasmati et al., 2011*), an anti-Tic110 antibody (*Aronsson et al., 2010*), an anti-Tic40 antibody (*Kasmati et al., 2011*), and an anti-green fluorescent protein antibody (Sigma, SAB4301138). In most cases, the secondary antibody used was goat anti-rabbit immunoglobulin G (IgG) conjugated with horseradish peroxidase (Sigma, 12-348); and protein bands were visualized via chemiluminescence using an ECL Plus western blotting detection kit (GE Healthcare) and an LAS-4000 imager (GE Healthcare). However, in the case of *Figure 5—figure supplement 1*, the secondary antibody was goat anti-rabbit IgG conjugated with alkaline phosphatase (Sigma, A3687), and the membrane was incubated with BCIP/NBT chromogenic substrate (Sigma, B3679).

The IP experiments were carried out as described previously (*Ling et al., 2019*). Constructs encoding YFP-conjugated fusion proteins (YFP-HA, SCE1-YFP, YFP-SUMO1, YFP-SUMO2, YFP-SUMO3) were transiently expressed in protoplasts. In some cases, the constructs were co-expressed with a construct encoding Toc33-HA. The protoplasts were solubilized using IP buffer containing 1% Triton X-100, and the resulting lysates were incubated with GFP-Trap beads (Chromotek). After four washes in IP buffer, the protein samples were eluted by boiling in SDS-PAGE loading buffer, and then analyzed by immunoblotting.

### Statistical analysis

The data from each experiment were analyzed in R. In most cases, two-tailed t-tests were performed. However, in one case, a one-way ANOVA was performed in conjunction with a Tukey HSD test (as indicated in the figure legend). The figures are annotated to indicate the level of significance, as follows: ns: not significant; *$p < 0.05$; **$p < 0.01$; ***$p < 0.001$; ****$p < 0.0001$; *****$p < 0.00001$.

### SUMO site prediction

The amino acid sequences of Toc159, Toc132, Toc120, Toc90, Toc75, Toc33, and Toc34 were retrieved from The *Arabidopsis* Information Resource (TAIR) website (*Berardini et al., 2015*). The GPS-SUMO algorithm was applied to all seven sequences (http://sumosp.biocuckoo.org/online.php; *Zhao et al., 2014*). The 'high stringency' setting was applied. The p-values were generated by the GPS-SUMO algorithm, and hits that were accompanied by p-values exceeding >0.05 were manually removed. The JASSA algorithm was also applied to all seven amino acid sequences (http://www.jassa.fr/index.php; *Beauclair et al., 2015*). In this case, the 'high cutoff' setting was applied. The GPS-SUMO and JASSA algorithms use fundamentally different methodologies (*Chang et al., 2018*).

## Acknowledgements

We are very grateful to Alistair Haslam for initiating the genetic experiments. We also thank Dr Errin Johnson and Raman Dhaliwal (Dunn School of Pathology, University of Oxford) for assistance with the electron microscopy, and Professor Jane Langdale for advice during the course of the work. We are grateful to Pedro Bota and Rita Ross for technical support. This work was supported by the Oxford Interdisciplinary Bioscience Doctoral Training Partnership (DTP), and by grants from the Biotechnology and Biological Sciences Research Council (BBSRC) (grant numbers BB/K018442/1, BB/N006372/1, BB/R016984/1, and BB/R009333/1) to RPJ.

## Additional information

### Competing interests

R Paul Jarvis: The application of CHLORAD as a technology for crop improvement is covered by a patent application (no. WO2019/171091 A).. The other authors declare that no competing interests exist.

## Funding

| Funder | Grant reference number | Author |
|---|---|---|
| Biotechnology and Biological Sciences Research Council | BB/K018442/1 BB/N006372/1 BB/R016984/1 BB/R009333/1 | R Paul Jarvis |
| Biotechnology and Biological Sciences Research Council | Interdisciplinary Bioscience Doctoral Training Partnership | Samuel Watson |

The funders had no role in study design, data collection and interpretation, or the decision to submit the work for publication.

## Author contributions

Samuel James Watson, Investigation, Methodology, Writing – original draft; Na Li, Yiting Ye, Investigation; Feijie Wu, Conceptualization, Investigation, Methodology; Qihua Ling, Conceptualization, Funding acquisition, Investigation, Methodology, Supervision, Writing – review and editing; R Paul Jarvis, Conceptualization, Funding acquisition, Investigation, Supervision, Writing – review and editing

## Author ORCIDs

Na Li (iD) http://orcid.org/0000-0002-3733-3373
Qihua Ling (iD) http://orcid.org/0000-0002-6984-9921
R Paul Jarvis (iD) http://orcid.org/0000-0003-2127-5671

## Decision letter and Author response

Decision letter https://doi.org/10.7554/eLife.60960.sa1
Author response https://doi.org/10.7554/eLife.60960.sa2

# Additional files

## Supplementary files

• Transparent reporting form

• Source data 1. Immunoblot source data. Uncropped immunoblot images for Figures 1 and 5, and their supplements.

• Source data 2. Numerical source data. Raw numerical data used to plot all graphs in the paper.

## Data availability

All data generated or analysed during this study are included in the manuscript and supporting files.

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
