## [Decision Letter]

**Acceptance summary:**

This paper is of interest to plant biologists and others studying the regulation of protein import into energetic organelles. The authors document suppression of chloroplast protein import mutants by SUMOylation mutants and provide evidence that chloroplast protein import machinery may be SUMOylated. Combined with their previous studies documenting ubiquitination of chloroplast protein import machinery, the authors propose that SUMOylation and ubiquitination may both play roles in regulating chloroplast protein import machinery.

**Decision letter after peer review:**

Thank you for submitting your article "Crosstalk between chloroplast protein import and the SUMO system revealed through genetic and molecular investigation" for consideration by *eLife*. Your article has been reviewed by 3 peer reviewers, one of whom is a member of our Board of Reviewing Editors, and the evaluation has been overseen by Christian Hardtke as the Senior Editor. The following individual involved in review of your submission has agreed to reveal their identity: Danny Schnell (Reviewer #2).

The reviewers have discussed the reviews with one another and the Reviewing Editor has drafted this decision to help you prepare a revised submission.

As the editors have judged that your manuscript is of interest, but as described below that additional experiments are required before it is published, we would like to draw your attention to changes in our revision policy that we have made in response to COVID-19 (https://elifesciences.org/articles/57162). First, because many researchers have temporarily lost access to the labs, we will give authors as much time as they need to submit revised manuscripts. We are also offering, if you choose, to post the manuscript to bioRxiv (if it is not already there) along with this decision letter and a formal designation that the manuscript is "in revision at eLife". Please let us know if you would like to pursue this option. (If your work is more suitable for medRxiv, you will need to post the preprint yourself, as the mechanisms for us to do so are still in development.)

Summary:

ppi1 mutants (defective in Toc33 component of the chloroplast protein import machinery) are small and yellow due to chloroplast defects. The authors have previously identified a suppressor of PPI1, a RING-type E3 ubiquitin ligase that can destabilize TOC components to regulate chloroplast protein import. Based on predicted SUMOylation sites, they speculated that the TOC complex might also be regulated by SUMOylation. Mild mutants in the single, essential Arabidopsis SUMO E2 conjugating enzyme (SCE1), one E3 ligase (SIZ1), or two of the three SUMO-encoding genes were able to partially suppress the chlorotic ppi1 mutant phenotype, implying that SUMOylation may also play a role in chloroplast protein import or chloroplast development. Similarly, SUMO overexpression enhanced the chlorosis of ppi1 mutants. The authors test for physical interactions between TOC components and SCE1 (the E2 conjugating enzyme, not SIZ1 the E3 ligase) using BiFC and Co-IP. They also document an interaction between the SUMOs and TOC159 via Co-IP, though this is strongest for SUMO3, which could not suppress the ppi1 phenotype. Finally, the authors detected high molecular weight bands when probing for TOC components after Co-IP for SUMO3-YFP, implying the SUMO3 maybe directly conjugated to TOC components. The authors propose a model in which SUMOylation of TOC proteins might negatively regulate their stability in the chloroplast outer membrane in combination with the ubiquitination mechanism that they have described previously.

Essential Revisions:

1) Although the changes in chlorophyll content and thylakoid morphology are consistent with impacts on protein import, they can be attributed to other processes that impact nuclear gene expression of chloroplast genes. The manuscript lacks a direct link between the very low levels of SUMOylation of Toc33 and an impact on protein import. In fact, the levels of the two nuclear encoded plastid proteins that they examine other than TOC components, Tic40 and Tic110, don't appear to change. Furthermore, the single immunoblot in figure 1I is the only evidence that addresses the impact of SUMOylation on TOC complex abundance or stability. Unfortunately, it does not provide conclusive evidence that the ppi1 siz1-4 double mutant stabilizes TOC complexes. Although it appears that the Toc75 band in the double mutant is slightly more intense, this could be attributed to variability in blotting, and the Toc159 band does not seem to increase significantly. This experiment would need to be repeated with quantification using replicates. Full blots will need to be presented in the supplemental data, including molecular weight markers and loading controls.

2) Similarly, Figure 5A, there are very faint bands for both TOC159 and 132 compared to the huge amount in the input and it is surprising that the authors see no flow through of SCE-1YFP. In 5B there is a band even in the YFP-HA control for Toc159 so how can the authors claim that Toc159 is indeed SUMOylated? This experiment would need to be repeated with quantification using replicates. Full blots will need to be presented in the supplemental data, including molecular weight markers and loading controls. Ideally, the authors would also generate a non-SUMO form of TOC159 and show that it does not bind SUMO.

3) Authors should characterize the T-DNA insertion alleles (sce1-4 and siz1-5) with respect to their effect on SUMO conjugation. This is crucial for the reviewer to determine if sce1-4 or siz1-4 has any significant role in global SUMOylation. To do this, authors should include data on the comparative SUMOylation immunoblot that includes WT, siz1-2/siz1-3, sce1-4 and siz1-5 and siz1-4. It will give the readers clarity about the role of these mutants in SUMO conjugation and how significantly different they are from previously validated E2 and E3 mutants.

4) The BiFC data presented in figure 3 is inconsistent and lacks appropriate controls. The C-termini of Toc33 and Toc34 are known to reside in the intermembrane space. If properly localized, we would not expect C-terminal cYFP fusions to exhibit BiFC with cytosolically localized SCE1nYFP fusions. Yet, the authors demonstrate BiFC and conclude that this supports their hypothesis for TOC SUMOylation. The similar topology is true for OEP7, which they use as a negative control. Furthermore, there is no clear evidence that the C-termini of Toc159 and Toc132 are in the cytosol, which also raises issues with these C-terminal fusions. On this basis, the authors cannot eliminate the possibility that the fusions are not properly localized to chloroplasts.

5) The authors speculate that SUMOylation works in concert or in opposition to the CHLORAD system. This group previously demonstrated that CHLORAD is important for remodeling TOC complexes during developmental events or stress responses. Do they propose something similar for SUMOylation? Since ppi1 is a full knockout mutant of the Toc33 isoform, how would SUMOylation defects complement a knockout? Do the authors propose that it is stability of the functionally redundant Toc34 isoform that is affected? If so, what is the link between Toc34 stabilization and Toc159 SUMOylation? Please discuss.

6) Additional controls are required for single/double mutant combinations:

a. sce1-4, siz1-4, and siz-15 single mutant analysis from Figure 2E/2F should be included in Figure 1G and H.

b. sum1-1, sum2-1, sum3-1, and sum1-1 sum2-1 mutants in Figure 4.

7) The n is very low and/or not clearly indicated for many of the plant phenotype experiments:

a. Please repeat chlorophyll content measurements in Figures 1 and 2, where the n is very low (3-7)

b. Please clearly indicate the exact n for each sample; this is often easier to do directly on the plots, rather than having a long list matching exact sample size to each genotype/treatment in the figure legend.

c. Given the small sample sizes for quantification in this study, please plot individual data points with summary statistics (mean/median, SD/SEM, or similar) overlaid. For larger n, a box plot is preferable to a bar plot.

8) RT-PCR is not ideal to test SUMO1 overexpression levels (Figure 4 supplement) – please use anti-SUMO antibody (as in Figure 5-S1) to evaluate protein levels in these lines.

[Editors' note: further revisions were suggested prior to acceptance, as described below.]

Thank you for resubmitting your work entitled "Crosstalk between chloroplast protein import and the SUMO system revealed through genetic and molecular investigation" for further consideration by *eLife*. Your revised article has been reviewed by 2 peer reviewers, one of whom is a member of our Board of Reviewing Editors, and the evaluation has been overseen by evaluated by Jürgen Kleine-Vehn (Senior Editor). The following individual involved in review of your submission has agreed to reveal their identity: Ari Sadanandom (Reviewer #3)

The manuscript has been improved but there are some remaining issues that need to be addressed, as outlined below:

Reviewer 1 has highlighted that either new experimental data should be added to directly assay protein import into the mitochondria or the text (including title, abstract, and main text) needs to be revised to clarify that the manuscript presents evidence that SUMOylation affects mitochondrial protein import machinery, rather than mitochondrial protein import.

Reviewer 3 has emphasized the need for studies on a non-SUMOylatable version of TOC159. However, it will also be possible to address these concerns through text edits throughout the manuscript. For example the authors can rephrase the last two sentences of the abstract to say, for example, that "the SUMO system plays a role in TOC protein stability" instead of using the phrase "SUMO system directly targets TOC proteins".*Reviewer #1:*

This paper is of interest to plant biologists and others studying the regulation of protein import into energetic organelles. The authors document suppression of chloroplast protein import mutants by SUMOylation mutants and provide evidence that chloroplast protein import machinery may be SUMOylated. Combined with their previous studies documenting ubiquitination during chloroplast protein import, the authors propose that SUMOylation and ubiquitination may both play roles in regulating chloroplast protein import machinery.

The authors have addressed many of the main reviewer comments in this revised version of the manuscript. Although there were some points they could not address due to pandemic-related restrictions, reasonable alternatives have been provided in most cases. However, I think that point 1 has not been sufficiently addressed. The authors have not documented any effects of SUMOylation on chloroplast protein import as claimed throughout the manuscript (e.g. last line of abstract). Rather, they have documented that SUMOylation affects chloroplast protein import machinery. This is an important distinction because, for example, the authors see no change in Tic110 import between the ppi1 single mutant and ppi1 combined sumoylation mutants (Figure 1I), suggesting that increased nuclear-encoded protein import is not the cause of ppi1 phenotype suppression in the double mutants between ppi1 and the SUMOylation pathway. This issue can be addressed by either 1) carefully rewording throughout the manuscript OR 2) Experientially assessing chloroplast protein import in ppi1 and double mutants between ppi1 and the SUMOylation pathway in more detail than Figure 1I.

*Reviewer #3:*

For the best proof for the role of SUMO in regulating Toc159 function in the Chlorad pathway the authors would need to show the impact of a non-SUMO version of Toc159. This is the standard in the SUMO field at the moment. Apart from this the authors have addressed my concerns adequately.

---

## [Author Response]

Essential revisions:1) Although the changes in chlorophyll content and thylakoid morphology are consistent with impacts on protein import, they can be attributed to other processes that impact nuclear gene expression of chloroplast genes. The manuscript lacks a direct link between the very low levels of SUMOylation of Toc33 and an impact on protein import. In fact, the levels of the two nuclear encoded plastid proteins that they examine other than TOC components, Tic40 and Tic110, don’t appear to change. Furthermore, the single immunoblot in figure 1I is the only evidence that addresses the impact of SUMOylation on TOC complex abundance or stability. Unfortunately, it does not provide conclusive evidence that the ppi1 siz1-4 double mutant stabilizes TOC complexes. Although it appears that the Toc75 band in the double mutant is slightly more intense, this could be attributed to variability in blotting, and the Toc159 band does not seem to increase significantly. This experiment would need to be repeated with quantification using replicates. Full blots will need to be presented in the supplemental data, including molecular weight markers and loading controls.

We agree that this is a very important point. To address this comment, we prepared fresh protein samples and repeated the immunoblotting analysis. Moreover, we included the *ppi1 siz1-5* genotype in the analysis (in addition to *ppi1 siz1-4*) to strengthen the experiment. Four independent, biological replicates were taken from plants of each genotype. The samples were taken from the batch of plants shown in Figure 1 – Supplement 1 on the day of photography. The new immunoblots show a clear increase in both Toc159 and Toc75 accumulation in both of the double mutants, relative to the *ppi1* control. Please see the new data in Figure 1I of the revised manuscript.

Tic110 was included as a compartment-specific loading control in this analysis. We quantified the Toc159 and Toc75 bands, and normalized the data using values for the Tic110 control. These new data show that both *ppi1 siz1-4* and *ppi1 siz1-5* display a statistically significant increase in the accumulation of both TOC proteins, relative to *ppi1*. Please see Figure 1J and Figure 1K.

We have included the full blots, including annotations and molecular weight markers in our revised manuscript. Please see Source data 1.

Lastly, we would like to clarify that Tic110 and Tic40 panels in the original figure served as loading controls, exactly as Tic110 does in the new experiment. The abundancies of these TIC proteins was not expected to change in the double mutants, as we expected that the SUMO system would exhibit specificity for TOC proteins in the same manner as the ubiquitin-based CHLORAD system (Ling et al., 2012; Ling et al., 2019).

2) Similarly, Figure 5A, there are very faint bands for both TOC159 and 132 compared to the huge amount in the input and it is surprising that the authors see no flow through of SCE-1YFP. In 5B there is a band even in the YFP-HA control for Toc159 so how can the authors claim that Toc159 is indeed SUMOylated? This experiment would need to be repeated with quantification using replicates. Full blots will need to be presented in the supplemental data, including molecular weight markers and loading controls. Ideally, the authors would also generate a non-SUMO form of TOC159 and show that it does not bind SUMO.

We thank the reviewers for making these important points, to which we have the following responses.

Firstly, regarding Figure 5A: SCE1 is a promiscuous enzyme that associates with thousands of proteins, and these interactions are highly transient. Therefore, we argue that it is actually not so surprising that the co-eluting Toc159 and Toc132 bands are quite faint; on the contrary, we think it is remarkable that we were able to detect even low-level co-elution in this experiment. To address this point, we have included a short additional statement in the manuscript (see Results). The anti-YFP beads used in this experiment are commercially produced and highly effective, and consequently it is not unexpected that there was no SCE1-YFP protein in the flow-through sample.

Secondly, regarding Figure 5B: We repeated this experiment and obtained a very similar result. Please see Figure 5 – Supplement 3. Moreover, we included quantification of the data in the new figure. We are fully confident in this result, especially as it is in agreement with results independently obtained by two other groups (Accossato et al., 2020; Elrouby and Coupland, 2010).

As requested, we have made the full blots, including annotations and molecular weight markers, available for each panel of Figure 5. Please see Source data 1. Unfortunately, our access to the laboratory has been severely restricted due to the ongoing pandemic, and as a result we were not able to create the non-SUMO form of Toc159.

3) Authors should characterize the T-DNA insertion alleles (sce1-4 and siz1-5) with respect to their effect on SUMO conjugation. This is crucial for the reviewer to determine if sce1-4 or siz1-4 has any significant role in global SUMOylation. To do this, authors should include data on the comparative SUMOylation immunoblot that includes WT, siz1-2/siz1-3, sce1-4 and siz1-5 and siz1-4. It will give the readers clarity about the role of these mutants in SUMO conjugation and how significantly different they are from previously validated E2 and E3 mutants.

We are grateful to the reviewers for making this is important point. To address this comment, we have thoroughly characterised the *siz1-4* and *siz1-5* mutant alleles (see below). However, as *sce1-4* has already been characterised (Saracco et al., 2007), and the mutant was found to show a moderate reduction in global SUMOylation, we do not believe it is necessary to repeat that analysis. Nonetheless, we have more clearly drawn attention to the prior published work in the revised text (see Results).

Regarding the *siz1* mutants: We carried out the suggested experiment, and found that *siz1-4* and *siz1-5* both show a decrease in global SUMOylation that is highly similar to the decrease observed in *siz1-2* and *siz1-3*, two previously characterised mutants. Please see Figure 1 – Supplement 3. The plants used in this analysis were subjected to heat shock to aid detection of SUMO1/2.

To further corroborate the usefulness of the *siz1-4* and *siz1-5* alleles, we have also included information on the respective T-DNA insertion sites, which both fall within the *SIZ1* open reading frame; please see Figure 1 – Supplement 2A. Moreover, we have added new data which confirm that both mutants show a major reduction in *SIZ1* expression; please see Figure 1 – Supplement 2B. Finally, we wish to emphasize that the *siz1-4* and *siz1-5* mutants also visually resemble *siz1-2* and *siz1-3*, and have added a statement to this effect in the revised manuscript (see Results).

Based on all of the aforementioned data, and given that SIZ1 has thousands of potential substrates and is one of the most important components of the SUMO pathway (Rytz et al., 2018), we are confident that the two new mutants show significant decreases in global SUMOylation.

4) The BiFC data presented in figure 3 is inconsistent and lacks appropriate controls. The C-termini of Toc33 and Toc34 are known to reside in the intermembrane space. If properly localized, we would not expect C-terminal cYFP fusions to exhibit BiFC with cytosolically localized SCE1nYFP fusions. Yet, the authors demonstrate BiFC and conclude that this supports their hypothesis for TOC SUMOylation. The similar topology is true for OEP7, which they use as a negative control. Furthermore, there is no clear evidence that the C-termini of Toc159 and Toc132 are in the cytosol, which also raises issues with these C-terminal fusions. On this basis, the authors cannot eliminate the possibility that the fusions are not properly localized to chloroplasts.

We thank the reviewers for bringing this point to our attention. Actually, the problem here was entirely related to our presentation, in the text and with the over-simplified annotation we employed in the relevant figure, and we sincerely apologize for the confusion that this has caused. We have reviewed all the constructs employed in this experiment, and can confirm that all of the TOC fusion proteins carried an N-terminal tag. The labelling in Figure 3 has now been changed to more clearly indicate that this is the case, and the relevant sections of text have been corrected. Please note that the OEP7 fusion protein carried a C-terminal tag, which is entirely in line with published information on the targeting properties of this protein (Lee et al., 2001).

5) The authors speculate that SUMOylation works in concert or in opposition to the CHLORAD system. This group previously demonstrated that CHLORAD is important for remodeling TOC complexes during developmental events or stress responses. Do they propose something similar for SUMOylation? Since ppi1 is a full knockout mutant of the Toc33 isoform, how would SUMOylation defects complement a knockout? Do the authors propose that it is stability of the functionally redundant Toc34 isoform that is affected? If so, what is the link between Toc34 stabilization and Toc159 SUMOylation? Please discuss.

Our model is that SUMOylation works in concert with the CHLORAD pathway. The two major components of the CHLORAD pathway – SP1 and SP2 – were originally identified as suppressors of *ppi1* in a genetic screen. It is therefore intriguing that a series of SUMO mutants seem to suppress *ppi1* in a similar fashion, and suggests that SUMOylation might promote CHLORAD function.

We have demonstrated that the *ppi1 siz1-4* and *ppi1 siz1-5* double mutants show clear increases in the accumulation of Toc75 and Toc159 (please refer to Figure 1I, 1J and 1K). These results suggest that SUMOylation acts to destabilise the TOC complex, and that, when its function is perturbed, TOC proteins accumulate at higher levels. We interpret that TOC complexes containing Toc34, Toc75 and Toc159 accumulate at higher levels in *ppi1 siz1-4* and *ppi1 siz1-5*, and that this synthetically improves the double mutant phenotypes relative to the *ppi1* control. We have added a statement to the text to more clearly outline our model (see discussion).

6) Additional controls are required for single/double mutant combinations:a. sce1-4, siz1-4, and siz-15 single mutant analysis from Figure 2E/2F should be included in Figure 1G and H.b. sum1-1, sum2-1, sum3-1, and sum1-1 sum2-1 mutants in Figure 4.

We agree that these controls will strengthen our experiments.

To address point (a), we repeated the *ppi1 sce1-4*, *ppi1 siz1-4*, and *ppi1 siz1-5* experiments alongside single mutant controls; please refer to Figure 1 – Supplement 1. As expected, the *sce1-4*, *siz1-4*, and *siz1-5* single mutants did not show an increase in chlorophyll abundance relative to wild-type plants, which underscores the synthetic nature of the effects observed in the *ppi1* double mutants.

To address point (b), we phenotypically characterised the *sum1-1* and *sum2-1* single mutants, as well as a *sum1-1*† *sum2-1* double mutant (where the dagger indicates that the line is heterozygous with respect to the mutation); please refer to Figure 4 – Supplement 1. We found that none of the mutants display a change in phenotype relative to wild type. Please note that we did not carry out this experiment alongside *ppi1*, *ppi1 sum1-1*, *ppi1 sum2-1*, and *ppi1 sum1-1*† *sum2-1* lines, as the experiment would otherwise have been too laborious, under the current challenging circumstances. This is because the *sum1-1*† *sum2-1* double mutants must be genotyped by PCR to identify *sum1-1* heterozygotes in a segregating population, and so it would have been a very large amount of work if all of the mutants were included in the experiment. We do not believe that this significantly affects interpretation of the results.

As for *sum3-1*, because we did not observe a change in phenotype in *ppi1 sum3-1* relative to the *ppi1* control, we respectfully argue that it is unnecessary to repeat the experiments Figures 4E and 4F alongside a *sum3-1* single mutant control.

7) The n is very low and/or not clearly indicated for many of the plant phenotype experiments:a. Please repeat chlorophyll content measurements in Figures 1 and 2, where the n is very low (3-7)b. Please clearly indicate the exact n for each sample; this is often easier to do directly on the plots, rather than having a long list matching exact sample size to each genotype/treatment in the figure legend.c. Given the small sample sizes for quantification in this study, please plot individual data points with summary statistics (mean/median, SD/SEM, or similar) overlaid. For larger n, a box plot is preferable to a bar plot.

To address point (a), we have repeated the chlorophyll content measurements in Figure 1B and 1H. In the repeated experiments, we also included *ppi1 siz1-5* to strengthen the conclusions, along with the single mutant controls that were requested in reviewer comment 6(a) above. However, we decided to also retain the original data in the manuscript (in Figure), with directions to the new experiments in the figure legends. The new data are shown in Figure 1 – Supplement 1. Unfortunately, we did not have enough seed remaining to repeat the experiments described in Figures 2B and 2D. However, in our experience, three or more biological replicates is usually sufficient in such analyses. Given that the results in these control experiments are clear, we hope the reviewers will forgive this small omission.

To address points (b) and (c), we remade every graph in the manuscript. All graphs now show all individual datapoints with a number above each bar to indicate the number of biological replicates per sample (wherever appropriate).

8) RT-PCR is not ideal to test SUMO1 overexpression levels (Figure 4 supplement) – please use anti-SUMO antibody (as in Figure 5-S1) to evaluate protein levels in these lines.

We attempted to grow plants to address this point, but unfortunately both *ppi1* SUMO1 OX lines showed poor germination rates, most likely due to the age of the seed and the severity of the mutant phenotypes. As a consequence of this we were unable to address this point thoroughly in a reasonable timeframe, and for this we apologise. However, we do feel that the RT-PCR results are clear, even if not ideal, and that they provide good evidence that the overexpressor lines were suitable for our analysis.[Editors’ note: further revisions were suggested prior to acceptance, as described below.]

The manuscript has been improved but there are some remaining issues that need to be addressed, as outlined below:Reviewer 1 has highlighted that either new experimental data should be added to directly assay protein import into the mitochondria or the text (including title, abstract, and main text) needs to be revised to clarify that the manuscript presents evidence that SUMOylation affects mitochondrial protein import machinery, rather than mitochondrial protein import.

To address this comment we have modified the Title, Abstract, and Main text as suggested.

More specifically:

– We edited the Title to make it more clear that the link we have identified is between the import machinery (system) and the SUMO system, as follows:

“Crosstalk between the chloroplast protein import and SUMO systems revealed through genetic and molecular investigation …”.

– For the same reason, we edited the last sentence of the Abstract, as follows:

“Thus, we have identified a regulatory link between the SUMO system and the chloroplast protein import machinery.”

– In the Introduction, we made changes to two sentence to address this point.

– In the Results, we made changes to two sentences to address this point.

– In the Discussion, we made changes to two sentences to address this point.

Reviewer 3 has emphasized the need for studies on a non-SUMOylatable version of TOC159. However, it will also be possible to address these concerns through text edits throughout the manuscript. For example the authors can rephrase the last two sentences of the abstract to say, for example, that "the SUMO system plays a role in TOC protein stability" instead of using the phrase "SUMO system directly targets TOC proteins".

To address this comment we have modified the Abstract and Main text as suggested.

More specifically:

– We edited the penultimate sentence of Abstract, as follows:

“Moreover, data from molecular and biochemical experiments support a model in which the SUMO system directly regulates TOC protein stability**.**”

– In the Results, we made changes to three sentences to address this point.

– In the Discussion, we made changes to ones sentence to address this point.

Reviewer #1:This paper is of interest to plant biologists and others studying the regulation of protein import into energetic organelles. The authors document suppression of chloroplast protein import mutants by SUMOylation mutants and provide evidence that chloroplast protein import machinery may be SUMOylated. Combined with their previous studies documenting ubiquitination during chloroplast protein import, the authors propose that SUMOylation and ubiquitination may both play roles in regulating chloroplast protein import machinery.The authors have addressed many of the main reviewer comments in this revised version of the manuscript. Although there were some points they could not address due to pandemic-related restrictions, reasonable alternatives have been provided in most cases. However, I think that point 1 has not been sufficiently addressed. The authors have not documented any effects of SUMOylation on chloroplast protein import as claimed throughout the manuscript (e.g. last line of abstract). Rather, they have documented that SUMOylation affects chloroplast protein import machinery. This is an important distinction because, for example, the authors see no change in Tic110 import between the ppi1 single mutant and ppi1 combined sumoylation mutants (Figure 1I), suggesting that increased nuclear-encoded protein import is not the cause of ppi1 phenotype suppression in the double mutants between ppi1 and the SUMOylation pathway. This issue can be addressed by either 1) carefully rewording throughout the manuscript OR 2) Experientially assessing chloroplast protein import in ppi1 and double mutants between ppi1 and the SUMOylation pathway in more detail than Figure 1I.

As detailed in our response to the first point of the Editors above, we have carefully addressed this point by making several changes to the text.

We wish to stress that the lack of an effect on Tic110 levels in the SUMO system mutants does not actually argue against our hypothesis that SUMO regulates chloroplast protein import. It is very well established that ubiquitination regulates the chloroplast protein import process via the CHLORAD system, and it is equally well established that CHLORAD affects the stability of *only* TOC proteins (i.e., TIC components, including Tic110, are not affected by CHLORAD) (Ling et al., 2012, 2019).

Reviewer #3:For the best proof for the role of SUMO in regulating Toc159 function in the Chlorad pathway the authors would need to show the impact of a non-SUMO version of Toc159. This is the standard in the SUMO field at the moment. Apart from this the authors have addressed my concerns adequately.

As detailed in our response to the second point of the Editors above, we have carefully addressed this point by making several changes to the text.

References

Accossato, S., Kessler, F., and Shanmugabalaji, V. (2020). SUMOylation contributes to proteostasis of the chloroplast protein import receptor TOC159 during early development. eLife 9, e60968.

Elrouby, N., and Coupland, G. (2010). Proteome-wide screens for small ubiquitin-like modifier (SUMO) substrates identify Arabidopsis proteins implicated in diverse biological processes. Proceedings of the National Academy of Sciences USA 107, 17415-17420.

Lee, Y.J., Kim, D.H., Kim, Y.W., and Hwang, I. (2001). Identification of a signal that distinguishes between the chloroplast outer envelope membrane and the endomembrane system in vivo. Plant Cell 13, 2175-2190.

Ling, Q., Huang, W., Baldwin, A. and Jarvis, P. (2012) Chloroplast biogenesis is regulated by direct action of the ubiquitin-proteasome system. Science 338, 655-659.

Ling, Q., Broad, W., Trösch, R., Töpel, M., Demiral Sert, T., Lymperopoulos, P., Baldwin, A. and Jarvis, R.P. (2019) Ubiquitin-dependent chloroplast-associated protein degradation in plants. Science 363, eaav4467.

Rytz, T.C., Miller, M.J., McLoughlin, F., Augustine, R.C., Marshall, R.S., Juan, Y.T., Charng, Y.Y., Scalf, M., Smith, L.M., and Vierstra, R.D. (2018). SUMOylome Profiling Reveals a Diverse Array of Nuclear Targets Modified by the SUMO Ligase SIZ1 during Heat Stress. Plant Cell 30, 1077-1099.

Saracco, S. A., Miller, M.J., Kurepa, J., and Vierstra, R.D. (2007). Genetic analysis of SUMOylation in Arabidopsis: conjugation of SUMO1 and SUMO2 to nuclear proteins is essential. Plant Physiology 145, 119-134.